# mTOR signaling regulates demand-adapted hematopoiesis and metabolic reprogramming required for an effective cellular immune response in *Drosophila melanogaster* larvae

Ines Anderl[1,2☯], Jens-Ola Ekström[1], Tea Tuomela[2], Mika Rämet[2], Tiina Susanna Salminen[2], Laura Vesala[1,2☯]*

**1** Department of Molecular Biology, Umeå University, Umeå, Sweden, **2** Faculty of Medicine and Health Technology, Tampere University, Tampere, Finland

☯ These authors contributed equally to the work
* laura.vesala@tuni.fi

## Abstract

The evolutionarily conserved mechanistic Target of Rapamycin (mTOR) pathway connects energy and nutrient availability to growth, proliferation, differentiation, immunity and survival. Here, we investigated the role of the mTOR pathway in *Drosophila* hematopoiesis and immunity using genetic and transcriptomic analyses of peripheral larval blood cells (hemocytes). We show that blood cell-directed *mTor* expression induced lamellocyte differentiation as seen after parasitoid wasp infection. Genetic epistasis revealed that lamellocyte hematopoiesis downstream of mTor is mediated by the JNK and p38 pathways. Transcriptomic profiling showed largely similar changes in gene expression patterns of wasp infected and *mTor* overexpressing hemocytes. While mTOR signaling is necessary for proper lamellocyte differentiation, mTOR Complex 1 (mTORC1) activity is suppressed in mature lamellocytes. Our transcriptome data indicated that hemocyte activation is accompanied by a shift in metabolism towards aerobic glycolysis for energy production, the oxidative pentose phosphate pathway for NADPH recycling, ROS production and detoxification as well as glutaminolysis for glutathione production. Our data highlight the key role of mTOR in controlling blood cell fate in *Drosophila*.

## Author summary

The immune system relies on specialized blood cells that can rapidly develop in response to infection or injury. However, the signals that guide blood cells to adopt specific immune functions are not fully understood. In this study, we used the fruit fly (*Drosophila melanogaster*), a widely used model organism that shares many conserved biological pathways with humans, to investigate how immune blood cells form during stress or infection. We focused on the mechanistic

**Data availability statement:** The transcriptomics data is available in Supporting information Table S1 and at GEO database with access number GSE237065. other relevant data are within the manuscript and its Supporting Information files.

**Funding:** This project was funded by the following grants: the Research council of Finland (276360) to LV, the Sigrid Jusélius foundation and Tampere Tuberculosis foundation to MR, Research council of Finland (322732, 328979 and 353367) and the Sigrid Jusélius foundation (3122800849) to TSS. The funders had no role in study design, data collection and analysis, decision to publish, or preparation of the manuscript.

**Competing interests:** The authors have declared that no competing interests exist.

Target of Rapamycin (mTOR) pathway, which is known to regulate cell growth and metabolism. We found that overexpressing *mTor* in circulating blood cells causes them to develop into lamellocytes, a specialized immune cell type normally produced during parasitic wasp infection. This response requires additional stress-related signaling pathways, suggesting that mTOR works together with immune signaling networks to control blood cell fate. Interestingly, while mTOR activity is necessary to initiate lamellocyte formation, its activity decreases once these cells mature. These findings identify mTOR as a key regulator of immune blood cell development in *Drosophila*.

## Introduction

All creatures great and small, men and flies alike, must fight infections to stay alive and healthy. The immune response is costly [1–4] because it requires metabolic support to provide energy and building blocks for the biosynthesis of cytokines and immune effector molecules as well as the generation and proliferation of appropriate immune cell types [5,6]. As such, the evolutionarily conserved mTOR (mechanistic Target of Rapamycin) network senses nutrient and energy status and coordinates these cues with changes in growth and metabolism [7], and in the presence of sufficient cellular energy and biomolecular components, mTOR mediates anabolic growth. mTOR is a serine/threonine kinase forming the catalytic subunit of two distinct protein complexes. As part of the mTOR Complex 1 (mTORC1), mTOR is associated with Raptor (Regulatory protein associated with mTOR) while it interacts with Rictor (Rapamycin-insensitive companion of mTOR) in the mTOR Complex 2 (mTORC2) [8]. mTORC1 controls cell growth and metabolism by promoting protein, lipid and nucleotide synthesis, by mediating the metabolic shift from oxidative phosphorylation (OXPHOS) to aerobic glycolysis and by inhibiting autophagy, lysosome biogenesis and apoptosis [9–12]. mTORC2 controls cytoskeletal organization, proliferation and cell survival [12]. It connects mTOR with insulin signaling by activating the AKT serine/threonine kinase (Akt), a core component in the insulin signaling pathway [13,14]. The Tuberous Sclerosis Complex (TSC), consisting of TSC1 and TSC2 (Tsc1 and Gigas in *Drosophila*), negatively regulates mTORC1, while the TSC is required for proper activation of mTORC2 [15,16]. TSC2 is a GTPase-activating protein inhibiting the GTP-bound form of Ras homology enriched in brain (Rheb) at the lysosome while TSC1 stabilizes TSC2 [16]. Due to its critical role in proliferation, differentiation and metabolism, mTOR has been intensively studied in hematopoiesis and immunity [17–20]. Data on mTOR in immune cell differentiation and function are dominated by studies of mammalian hematopoietic stem cells (HSCs) and T cells. In HSCs, mTORC1 controls self-renewal and differentiation [21,22]. In steady state T cells, the activity of mTOR is kept at bay by multiple inhibitory mechanisms to maintain T cell homeostasis. Following immune activation, mTOR orchestrates the metabolic changes leading to T cell activation and differentiation of T cell lineages [18,23]. Moreover, B cells and innate immune cells also rely on mTOR signaling for activation and differentiation [19,24].

In the fruit fly, *Drosophila melanogaster*, blood cells, also known as hemocytes, mediate the cellular immune response [25–29]. Plasmatocytes, initially described as phagocytes, are the most abundant hemocyte type in homeostasis. They are highly plastic and are able to transdifferentiate into crystal cells and lamellocytes [30–33]. Different subsets of plasmatocytes may be involved in a wider variety of specific functions [34–39], revealing a more complex blood cell system than previously anticipated. Crystal cells produce melanin and participate in the melanization of bacterial pathogens and parasitoids [40,41]. Parasitoid wasps, such as *Leptopilina boulardi*, trigger the hematopoiesis of lamellocytes, specialized immune cells encapsulating parasitoids [42,43]. Several signaling cascades are implicated in lamellocyte hematopoiesis such as the JAK-STAT, JNK and p38 pathways [44,45]. Despite the importance of mTOR signaling in mammalian hematopoiesis and immunity, little is known of its role in *Drosophila* hemocytes. So far, it has only been studied in the context of lymph gland and hematopoietic stem cell development where the activation of mTOR signaling increases proliferation and growth of the posterior signaling center [46] and promotes hemocyte progenitor maintenance in the medullary zone [47]. Here, we investigate mTOR in homeostasis and after parasitoid wasp infection in the peripheral hemocytes of *D. melanogaster* larvae by utilizing genetics and transcriptome analysis. Our data highlight the key role of *Drosophila* mTOR signaling in controlling immune cell fate decisions.

## Results

### *mTor^WT* overexpression in hemocytes causes melanotic nodule formation and improves the encapsulation response against the parasitoid wasp Leptopilina boulardi G486

To study the effects of *mTor*, we expressed a wild-type *mTor* construct (*UAS-mTor^WT* [48] in larval hemocytes with the combined *Hml^Δ-He*-GAL4 driver (hereafter *HH>* [33]). *mTor^WT* overexpression in hemocytes (*HH>mTor^WT*) induced melanotic nodule formation mainly in the posterior ends of the larvae (Fig 1A-A'", white triangles). As expected, we detected much higher *mTor* mRNA levels in *HH>mTor^WT* hemocytes compared to controls. Surprisingly, hemocytes of *mTor^WT* overexpressing larvae without nodules (*HH>mTor^WT*-) expressed significantly higher levels of *mTor* than those with nodules (*HH>mTor^WT*+; Fig 1B). Because melanotic nodules are reminiscent of the melanized capsules hemocytes form around parasitoid wasp eggs, we exposed larvae to *L. boulardi* to test whether *mTor* was required for the encapsulation response against parasitoid wasps. We expressed a 754 amino acid central region known as the "*Toxic Effector Domain*" (*Tor^TED*, hereafter *mTor^DN*), which functions as a dominant negative by sequestering factors essential for mTOR signaling [48]. 48 hours after infection, *mTor^WT* increased while *mTor^DN* decreased wasp killing revealing that mTor is involved in the encapsulation response (Fig 1C).

### *mTor^WT* overexpression causes hemocyte activation similar to parasitoid wasp infection, while suppression of *mTor* affects mainly plasmatocyte lineage hemocytes

After activation by parasitoid wasp infection, hemocytes undergo several changes: *i.* total hemocyte counts double; *ii.* steady state plasmatocyte numbers decrease in favor of activated plasmatocytes and *iii.* lamelloblasts appear and give rise to lamellocytes via an intermediate cell type named prelamellocytes [33]. To investigate whether *mTor^WT* overexpression activates blood cells in a similar manner as wasp infection, we used the blood cell reporter *Me* together with the *HH* driver (*MeHH>* [33]). The total and lineage-specific blood cell counts of uninfected *MeHH>mTor^WT* larvae closely mimicked the wasp infection pattern described above (Figs 1D-D"; S1A). The total number of hemocytes increased after *mTor^WT* overexpression (Fig 1D), with all subclasses expanding except for the basal plasmatocytes, whose numbers decreased (Fig 1D'-D"). Because fewer than half of *MeHH>mTor^WT* larvae had melanotic nodules, we investigated if the total and lineage blood cell counts differed between larvae without nodules (*MeHH>mTor^WT*-) and those with nodules (*MeHH>mTor^WT*+). We found that nodules were correlated with a higher degree of hemocyte activation (S2A-B" Fig). By contrast, when we suppressed *mTor* in wild type larvae (*MeHH>mTor^DN*), total hemocyte counts were significantly reduced due to the reduction of plasmatocytes (Fig 1D). No activated plasmatocytes or lamellocytes were produced (Figs 1D'-D"; S1A). Similarly, *mTor* RNAi

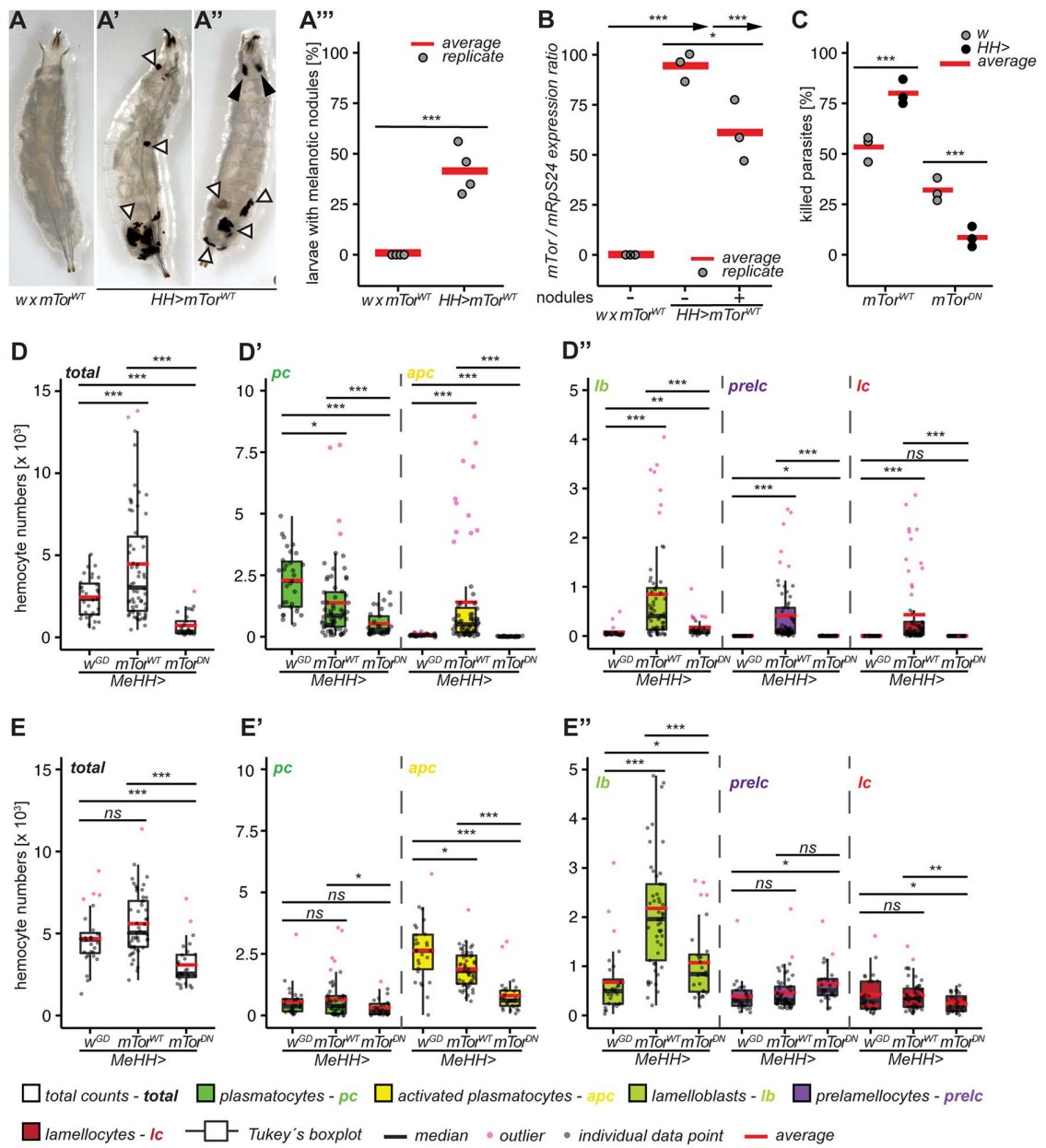

**Fig 1. *mTor* expression in hemocytes induces melanotic nodules and is required for the encapsulation response as well as hemocyte activation.** A) Melanotic nodule phenotype of control larvae (*w x mTor^WT*, n = 136); A'-A") Melanotic nodule phenotype of *mTor^WT* overexpression larvae (*HH>mTor^WT*, n = 266, white triangles - nodules, black arrowheads - salivary glands); A'") Penetrance of the melanotic nodule phenotype in control and *mTor^WT* overexpressing larvae. We also detected hypotrophy and melanization of salivary glands (Fig 1A," black arrowheads) presumably due to ectopic *He-GAL4* expression in this tissue. B) *mTor/mRpS24* expression ratio of control (*w x mTor^WT*) and *mTor^WT* overexpressing hemocytes without (*HH>mTor^WT*-) and with (*HH>mTor^WT*+) melanotic nodules, arrows represent comparison to control. C) Parasitoid killing assay of *w x mTor^WT* (n = 313), *HH>mTor^WT* (n = 305), *w x mTor^DN*, (n = 147) and *HH>mTor^DN* (n = 153). D) Total hemocyte counts of uninfected control (*MeHH>w^GD*, n = 31), *mTor^WT* overexpression (*MeHH>mTor^WT*, n = 64 and loss of *mTor* function (*MeHH>mTor^DN*, n = 29) larvae. D') Cell counts of plasmatocyte lineage; D") Cell counts of lamellocyte lineage. E) Total hemocyte counts of infected control (*MeHH>w^GD*, n = 28), *mTor^WT* overexpression (*MeHH>mTor^WT*, n = 31) and loss of *mTor* function (*MeHH>mTor^DN*, n = 28) larvae; E') Cell counts of plasmatocyte lineage; E") Cell counts of lamellocyte lineage. Significance levels: \*\*\* $p < 0.0001$, \*\* $p < 0.001$, \* $p < 0.05$, ns – not significant. The hemocyte count and wasp encapsulation data are available in the S7 Table.

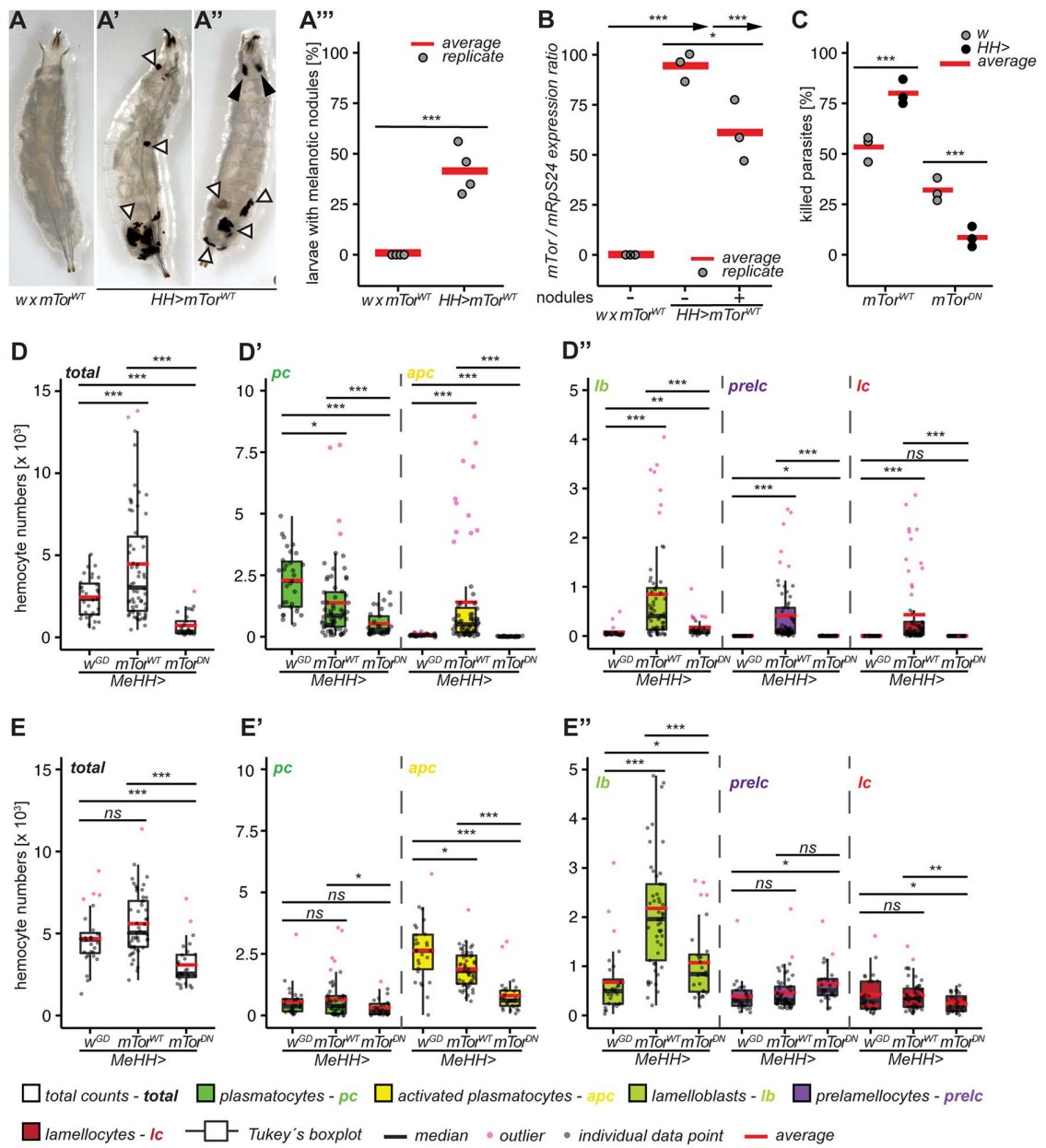

(*mTor^IR*) in hemocytes reduced plasmatocyte counts and did not lead to production of activated plasmatocytes nor lamellocytes. Of note, *mTor^IR* increased the lamelloblast counts to a larger extent than *mTor^DN* (S3A-A" Fig).

Two days after wasp infection, control (*w^GD*) larvae had increased numbers of activated plasmatocytes and lamellocyte lineage hemocytes (compare Figs 1E-E" to 1D-D"; S1A). These effects were only marginally further affected by *mTor^WT* overexpression, except that wasp-infected *MeHH>mTor^WT* larvae had more lamelloblasts than the wasp-infected *w^GD* control (Fig 1E'-E"). In contrast, *MeHH>mTor^DN* larvae had significantly lower total hemocyte counts (Fig 1E) mostly due to decreased number of activated plasmatocytes (Fig 1E'). However, only small effects were seen on the precursor cells of the lamellocyte lineage: lamelloblasts and prelamellocytes increased, but fewer lamellocytes were formed (Fig 1E"). *mTor* RNAi (*mTor^IR*) phenocopied the *mTor^DN* hemocyte phenotype except for decrease in total hemocyte and lamellocyte counts (S3B-B" Fig). Thus, our data show that *mTor^WT* overexpression induces and phenocopies the cellular immune response to parasitoid wasp infection and that mTor is required for a proper encapsulation response. *mTor^DN* and *mTor^IR* accumulate precursor lamelloblasts and reduce plasmatocyte lineage hemocytes under homeostasis and after infection, and *mTor^DN* partially blocks lamellocyte hematopoiesis. Therefore, *mTor* has a central role in regulating hemocyte proliferation and differentiation.

### Releasing the negative regulation of mTORC1 by silencing the TSC does not fully phenocopy *mTor^WT* overexpression

Mutations in either component of the Tuberous Sclerosis Complex (TSC) release the negative regulation on mTORC1 [49]. We used RNAi to test if silencing *Tsc1* or *gigas* (*gig*) mimic the *mTor^WT* overexpression phenotype. In uninfected larvae, silencing either component of the TSC did not phenocopy the *mTor^WT* effect on lamellocytes. However, the effect on the plasmatocyte lineage was similar (decreased plasmatocytes and increased activated plasmatocyte numbers S4A-A," S5A-A"Figs). After wasp infection, both knockdowns increased lamelloblasts, and T*sc1* RNAi reduced plasmatocyte lineage counts (S4A-B," S5B-B"Figs). Furthermore, *gig* RNAi enhanced the killing of wasp larvae (S5C Fig), whereas killing was either unaffected or slightly reduced by *Tsc1* RNAi (S4C Fig). Thus, silencing of *Tsc1* or *gig* is not sufficient to fully phenocopy *mTor^WT*. Next, we opted for a combined suppression of *Tsc1* and *gig*. The double knockdown increased plasmatocyte, activated plasmatocyte and lamelloblast numbers, while lamellocytes were only marginally increased (Figs 2A-A"; S1B). However, neither the hemocyte composition of wasp-parasitized larvae (Figs 2B-B"; S1B) nor the response against wasps (Fig 2C) were different from controls. Taken together, silencing the TSC did not induce lamellocyte hematopoiesis comparable to *mTor^WT* overexpression. However, it affected plasmatocyte-like hemocytes (plasmatocytes, activated plasmatocytes and lamelloblasts) underscoring the role of mTORC1 in the proliferation of these cell types.

### mTORC1 and mTORC2 are both required for hemocyte activation and the immune response against *L. boulardi* infection

Because *mTor^WT* overexpression and TSC silencing had somewhat distinct phenotypes, we studied the discrete roles of mTORC1 and mTORC2 by individually silencing the adaptor proteins, *raptor* and *rictor*. *Raptor* RNAi strongly increased lamelloblasts, while *rictor* RNAi increased plasmatocytes (S6A-A" Fig). After parasitization, *raptor* knockdown led to increased lamelloblasts and prelamellocyte numbers, while *rictor* knockdown had only a minor effect on lamelloblasts (S6B-B" Fig). The wasp assay confirmed that neither *raptor* nor *rictor* silencing alone was sufficient to impede the encapsulation response (S6C Fig).

Because mTORC1 and mTORC2 both require mTor, we investigated the effect of combined suppression of *raptor* and *rictor*. As for *raptor* RNAi alone, only lamelloblast counts were affected in unchallenged double knockdown larvae (Figs 3A-A"; S1C). However, both the encapsulation response (Fig 3B) and lamellocyte numbers after wasp infection (Figs 3C-C"; S1C) were significantly reduced. In conclusion, both endogenous mTORC1 and mTORC2 are required for inducing a full cellular immune response including lamellocyte hematopoiesis (summarized in S7 Fig).

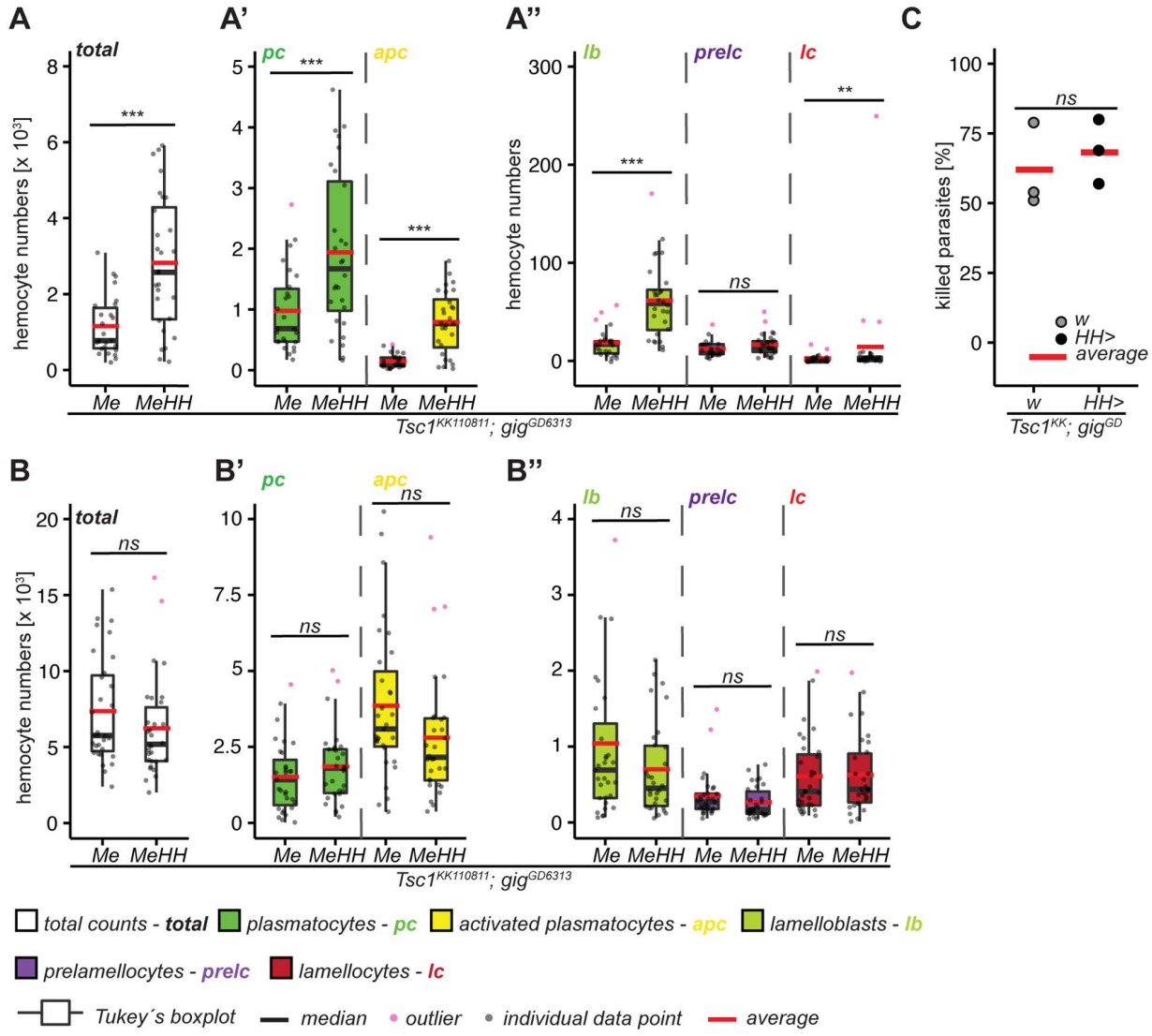

**Fig 2. *Tsc1-gig* double knock-down increases plasmatocyte lineage and lamelloblasts but does not activate lamellocyte hematopoiesis. A)** Total hemocyte counts (n = 28-31) of uninfected control *Me > Tsc1^KK110811^; gig^GD6313^* and *Tsc1-gig* double knock-down (*MeHH > Tsc1^KK110811^; gig^GD6313^*) larvae; A') Cell counts of plasmatocyte lineage of uninfected larvae; A") Cell counts of lamellocyte lineage of uninfected larvae. **B)** Total hemocyte counts of infected larvae; B') Cell counts of plasmatocyte lineage of infected larvae; B") Cell counts of lamellocyte lineage of infected larvae. **C)** Parasitoid wasp infection assay of control (w x *Tsc1^KK^; gig^GD^*, n = 155) and *Tsc1-gigas* double knock down (*HH > Tsc1^KK^; gig^GD^*, n = 151) in hemocytes. Significance levels: *** *p < 0.0001*, ** *p < 0.001*, * *p < 0.05*, *ns* – not significant. The hemocyte count and wasp encapsulation data are available in the S7 Table.

### *mTor^WT^* and parasitoid wasp infection induce largely overlapping changes in hemocyte gene expression

Because of the similarities between the *mTor^WT^* overexpression and parasitoid wasp-induced hemocyte phenotypes (Fig 4A), we asked whether they regulate the expression of similar sets of genes, and whether mTORC1- and mTORC2-related gene expression patterns are affected. We performed a bulk transcriptome analysis of hemocytes from *HH > mTor^WT^* and control *HH x w^GD^* larvae, both without and with *L. boulardi* infection. A principal component analysis showed that hemocytes of *mTor^WT^* overexpression and infected control larvae separated from uninfected controls along the PC1 and PC2 axes and from each other in the PC2 dimension. Hemocytes from uninfected and infected *mTor^WT^*

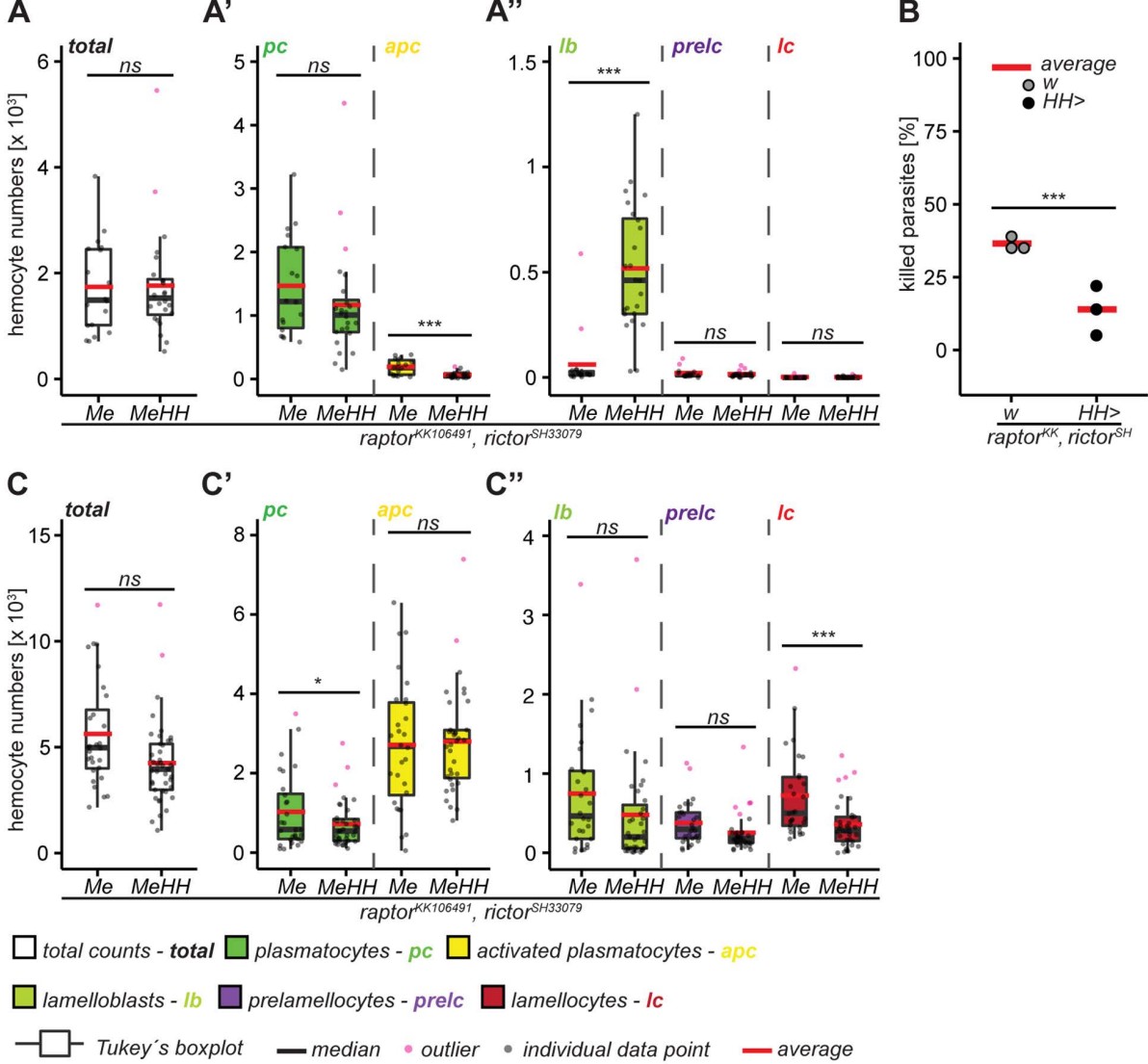

Fig 3. **Raptor-rictor** double knockdown weakens the immune response and reduces lamellocytes after infection. **A)** Total hemocyte counts of uninfected control (*Me x raptor*[KK106491],*rictor*[SH33079], n = 18) and *raptor-rictor* double knockdown (*MeHH>raptor*[KK106491],*rictor*[SH33079], n = 25); A') Cell counts of plasmatocyte lineage; A") Cell counts of lamellocyte lineage. **B)** Parasitoid wasp assay of control (*w x raptor*[KK106491],*rictor*[SH33079]; n = 159) and *raptor-rictor* double knockdown (*HH>raptor*[KK106491],*rictor* [SH33079]; n = 137). **C)** Total hemocyte counts of infected control (*Me x raptor*[KK106491],*rictor*[SH33079]) and *raptor-rictor* double knock-down (*MeHH>raptor*[KK106491],*rictor*[SH33079]); C') Cell counts of plasmatocyte lineage; C") Cell counts of lamellocyte lineage. Significance levels: *** $p < 0.0001$, ** $p < 0.001$, * $p < 0.05$, *ns* – not significant. The hemocyte count and wasp encapsulation data are available in the S7 Table.

overexpression clustered together (Fig 4B). Thus, we focused our analyses on data from *mTor*[WT] overexpression and infected control hemocytes (S1 Table). Many genes were significantly affected in the same direction by both treatments (889 up- and 716 downregulated; Fig 4C, red dots). The number of genes affected by *mTor*[WT] overexpression only (471 up- and 765 downregulated; Fig 4C, green dots) was higher than those specific for infection (124 up-and 102 downreg- ulated; Fig 4C, blue dots). It should be noted that our bulk RNA-sequencing approach does not permit the discrimination of transcriptomic profiles at the level of individual hemocyte subtypes. Therefore, the data represent an aggregate over- view of collective transcriptional changes in response to the treatments. Nevertheless, single cell sequencing signatures

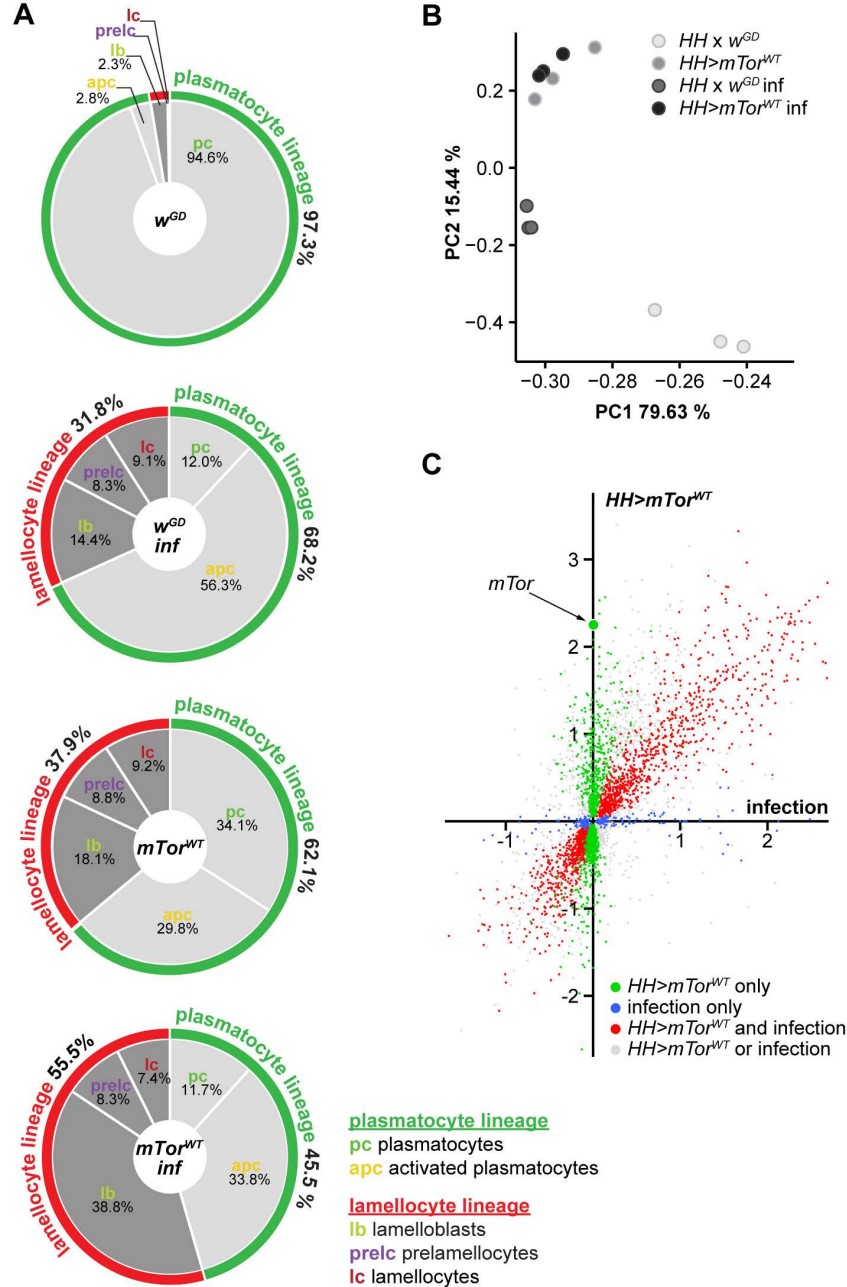

**Fig 4. *mTor^WT* and parasitoid wasp infection induce largely overlapping changes in hemocyte gene expression.** A) Percentages of plasmatocyte (green) and lamellocyte (red) lineage hemocytes of uninfected controls (*MeHH>w^GD*), wasp infected controls (*MeHH>w^GD inf*), *mTor^WT* overexpression (*MeHH>mTor^WT*) and wasp infected *mTor^WT* overexpression (*MeHH>mTor^WT inf*) larvae. *mTor^WT* and wasp infection induced very similar hemocyte profiles, but wasp infection induced a higher activated-plasmatocyte-to-plasmatocyte ratio. Moreover, *mTor^WT inf* had increased lamelloblasts, while all other hemocyte populations were comparable to wasp infection. Percentages are based on the hemocyte analysis shown in Fig 1D-E." B) Principal component analysis of RNAseq data of hemocytes from uninfected controls (*HH×w^GD*), wasp infected controls (*HH×w^GD inf*), *mTor^WT* overexpression (*HH>mTor^WT*) and wasp infected *mTor^WT* overexpression (*HH>mTor^WT inf*) larvae. C) Relative gene expression of late third instar *mTor^WT*-overexpressing hemocytes and hemocytes of wasp infected larvae 48 hours post infection. Only the 5350 significantly affected genes are shown. Each point represents the average of three replicates. Genes significantly affected by *mTor^WT* overexpression only are in green and those affected by *infection only* in blue. Genes affected by both experimental conditions are shown in red. The criterium for a significant effect was set to < 15% FDR and for the absence of a significant effect to > 50% FDR. Genes that are significantly affected in one treatment while the effect of the other is uncertain (15–50% FDR) are shown in grey. The gene expression data are available in the S1 Table.

PLOS Genetics

of hemocytes can be used to infer gene expression patterns of hemocyte subtypes within our bulk RNAseq dataset. The similarities in the gene expression patterns were likely attributed to the expression patterns of the prevalent hemocyte types present in both wasp-infected and *mTor^WT^* overexpression samples, i.e., lamellocyte lineage hemocytes and activated plasmatocytes (Fig 4A). Also, the expression patterns of the consensus markers for plasmatocytes, lamellocytes and crystal cells ([29]; S1 Table) were similar between *mTor^WT^* overexpression and wasp infection. Exceptions were a few crystal cell markers, *Hsp83*, *GstE3*, *lazaro* and *CG34136* which were all preferentially or exclusively upregulated after *mTor^WT^* overexpression (S8A Fig). The general trend was upregulation of lamellocyte markers and downregulation of crystal cell- and plasmatocyte-associated genes (S8A Fig). Nevertheless, some plasmatocyte subclusters were differentially regulated. The subcluster "stress/GST" was specifically upregulated by *mTor^WT^* overexpression, while "mitotic" was downregulated. The "AMP" subcluster was only upregulated after wasp infection (S8B Fig) suggesting that additional signals, such as signaling molecules and bacterial contamination brought on by the oviposition, elicit the humoral immune response. GO term analysis showed downregulation of terms related to mitosis and cell cycle by *mTor^WT^* overexpression and enrichment of genes connected to antibacterial humoral response (S8C Fig, S2 Table) for wasp infection. GO terms associated with changes in cytoskeleton, cell shape and cell communication and signaling were enriched for genes upregulated by both wasp infection and *mTor^WT^* overexpression and largely overlap with the set of lamellocyte-specific genes [29]. GO terms connected to anabolic growth were downregulated by *mTor^WT^* overexpression and wasp infection. Moreover, the similarities of the gene expression patterns of naturally (wasp infection) versus genetically induced (*mTor^WT^*) hemocyte activation suggest that similar mechanisms are required during the activation process.

## mTORC1-related gene expression is downregulated after wasp infection and *mTor^WT^* overexpression

Next, we scrutinized our transcriptome data for mTOR pathway genes, mTOR core components (S9B Fig), genes activating the mTOR pathway (S9A; B'-B''' Fig), negative regulators of mTORC1 (S10A-A'' Fig) and biosynthetic pathways activated by mTOR (S11 Fig). We did not observe major transcriptional regulation of the mTOR complex components in either treatment (S9B Fig). mTORC1 is activated by amino acids and growth factor signaling, while mTORC2 is induced by insulin-like signaling but also responds to growth factors [12]. Many genes in the amino acid sensing pathway were downregulated (S9B' Fig), while growth factor (S9B'' Fig) and insulin-like signaling (S9B''' Fig) were induced in activated hemocytes. mTOR promotes anabolic growth through its function as a serine threonine kinase within mTORC1 and by acting as a transcription factor or transcriptional coactivator [50]. However, biosynthetic pathways and energy metabolism normally stimulated by mTORC1 such as protein and nucleotide synthesis as well as oxidative phosphorylation were predominantly downregulated after wasp infection and *mTor^WT^* overexpression (Fig 5, S3 Table). Furthermore, the transcription factors promoting these anabolic growth pathways such as *Myc*, *Sterol regulatory element binding protein* (*SREBP*), *similar* (*sima*), *Hif-1α/tango* (*tgo*, *Hif-1ß*) and *spargel* (*srl*) as well as the key enzymes regulating nucleotide synthesis such as *rudimentary* (*r*), *Serine hydroxymethyl transferase* (*Shmt*) and *NAD-dependent methylenetetrahydrofolate dehydrogenase* (*Nmdmc*) were downregulated or not affected (S11A Fig). Processes normally suppressed by mTORC1 such as autophagy, lysosome biogenesis and apoptosis (S11A' Fig) were not explicitly suppressed in our data except for downregulation of all *Vacuolar-type ATPase* (*V-ATPase*) genes in *mTor^WT^* hemocytes. On the contrary, many apoptotic, lysosomal and autophagic genes were upregulated. Stress-related Unfolded Protein Response (UPR) genes were upregulated mainly by *mTor^WT^* overexpression (Fig 5, S3 Table [50]). The negative regulators of mTORC1, AMPK, the *Regulated in development and DNA damage response 1* (*REDD1*) [51] ortholog scylla (*scyl*) [52] and p53 [53] were upregulated and the negative regulators of p53, *Companion of reaper* (*Corp*) [54] and *Septin interacting protein 3* (*Sip3*) [55] were downregulated (S10A'-A'' Fig). However, the changes in gene expression levels were only modest and might not directly correspond with mTORC1 inhibition at the protein level. Alternatively, mTORC1 suppression may be hemocyte subtype-specific, yet potentially concealed in bulk sequencing data. Although mTOR signaling is required for successful wasp defense, hemocyte activation and differentiation, the hemocyte gene expression patterns after wasp parasitization and *mTor^WT^* overexpression

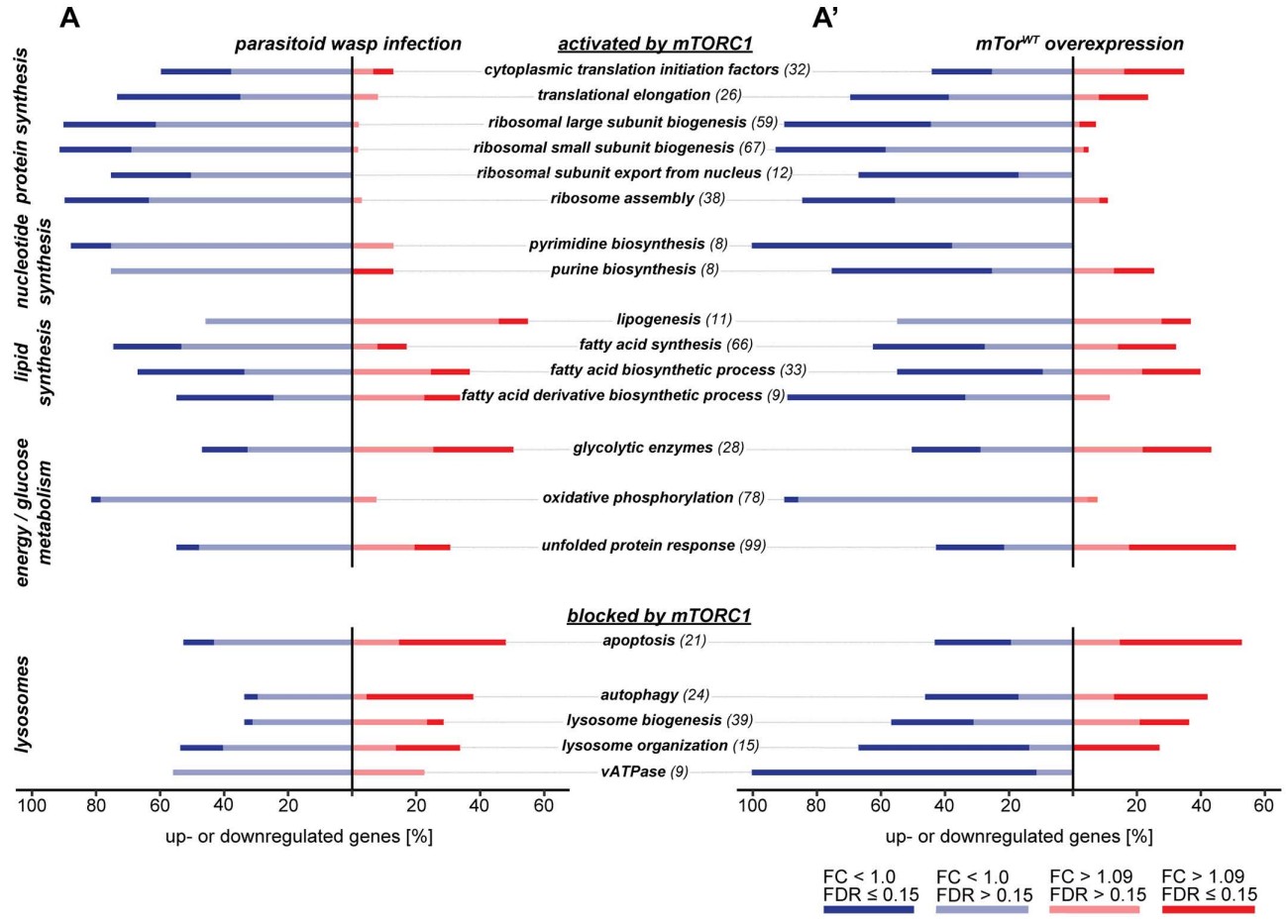

**Fig 5. Expression of gene groups involved in anabolic and catabolic processes regulated by mTORC1.** A) Gene groups associated with anabolic processes promoted by mTORC1-mediated phosphorylation (protein synthesis, nucleotide synthesis, lipid synthesis, glucose metabolism, energy metabolism) and unfolded protein response; A') Gene groups associated with catabolic processes blocked by mTORC1-mediated phosphorylation (apoptosis, autophagy, lysosome biogenesis). The gene expression data are available in the S3 Table.

did not strongly indicate activated mTOR signaling. Rather, they indicate that mTORC1 signaling may be downregulated in lamellocytes.

### Translation is inhibited in lamellocytes after *mTor^WT^* overexpression and wasp infection

To further explore the potential inhibition of mTORC1 in lamellocytes, we focused on the regulation of translation initiation, one of the most well-known conserved outputs of mTORC1 [56], as a read-out for mTORC1 activity. Translation initiation is activated by S6k [57] and suppressed by Thor, the *Drosophila* homolog of the Eukaryotic translation initiation factor 4E binding protein (4E-BP), which inhibits the eukaryotic translation initiation factor 4E1 (eIF4E1) activity [58,59]. *S6k* was significantly upregulated after *mTor^WT^* overexpression and after parasitization. However, *eIF4E1* was significantly downregulated by both treatments and, moreover, *Thor* was upregulated by *mTor^WT^* overexpression (S11A Fig). To further investigate if translation is suppressed after wasp infection or by *mTor^WT^* overexpression, we performed a western blot of total S6k and pS6k from bulk hemocyte samples. The pS6k/S6k ratio did not significantly differ between control, *mTor^WT^* and wasp-infected hemocytes (S12A-B' Fig) suggesting that S6k activity was not increased by *mTor^WT^* overexpression

or wasp infection. This might be due to the suppression of mTORC1 activity in a specific hemocyte type. Therefore, we stained hemocytes with an antibody against the phosphorylated ribosomal protein S6 (pRpS6; at S6k-specific activating phosphorylation sites $S_{233/235/239}$; S13A Fig). The amount of pRpS6 protein varied between treatments and was lowest in *mTor^WT* overexpressing hemocytes (S13B-C Figs). However, regardless of treatment, lamellocytes had no or very dim pRpS6 staining compared to other hemocyte types (Figs 6A-D; S13B) indicating that mTORC1-mediated translation, and perhaps mTORC1 activity in general, is suppressed in lamellocytes but remains active in plasmatocyte subtypes.

### Hemocyte activation by *mTor^WT* overexpression and wasp infection leads to changes in genes involved in glucose and glutamine metabolism

During hematopoiesis and immune activation, hemocytes undergo profound changes in their transcriptional and metabolic profiles [2,60–63]. To get further insights into the gene expression patterns of metabolic genes in hemocytes induced by natural parasitoid wasp infection and overexpression of *mTor^WT*, we assayed the transcripts of the central metabolic pathways (see S4 Table for metabolic gene expression). *Glucose metabolism* - Trehalose and glucose transporters as well as *Trehalase* (*Treh*) were among the most upregulated genes in both treatments (Fig 7A-B). *Tret1–1*, *CG1208* and *Treh* were established as lamellocyte markers [29], and recently, an *in vivo* reporter confirmed the lamellocyte-specific expression of *Treh* [63]. The increased expression of sugar transporters implies an increased flow through glycolysis. The upregulation of *Lactate dehydrogenase* (*Ldh*) and the downregulation of *Pyruvate dehydrogenase kinase* (*Pdk*) further indicate a shift towards aerobic glycolysis (Fig 7A, C). The rate-limiting enzyme of the oxidative pentose phosphate pathway (PPP), *Zwischenferment* (*Zw*), was significantly upregulated under both conditions, while the expression of *Transaldolase* (*Taldo*), the main enzyme of the non-oxidative PPP, was not affected (Fig 7D-D'). Zw oxidizes glucose-6-phosphate (G6P), thereby generating NADPH in the oxidative branch of the PPP (Fig 7A). The other NADPH-producing enzymes *Phosphogluconate*

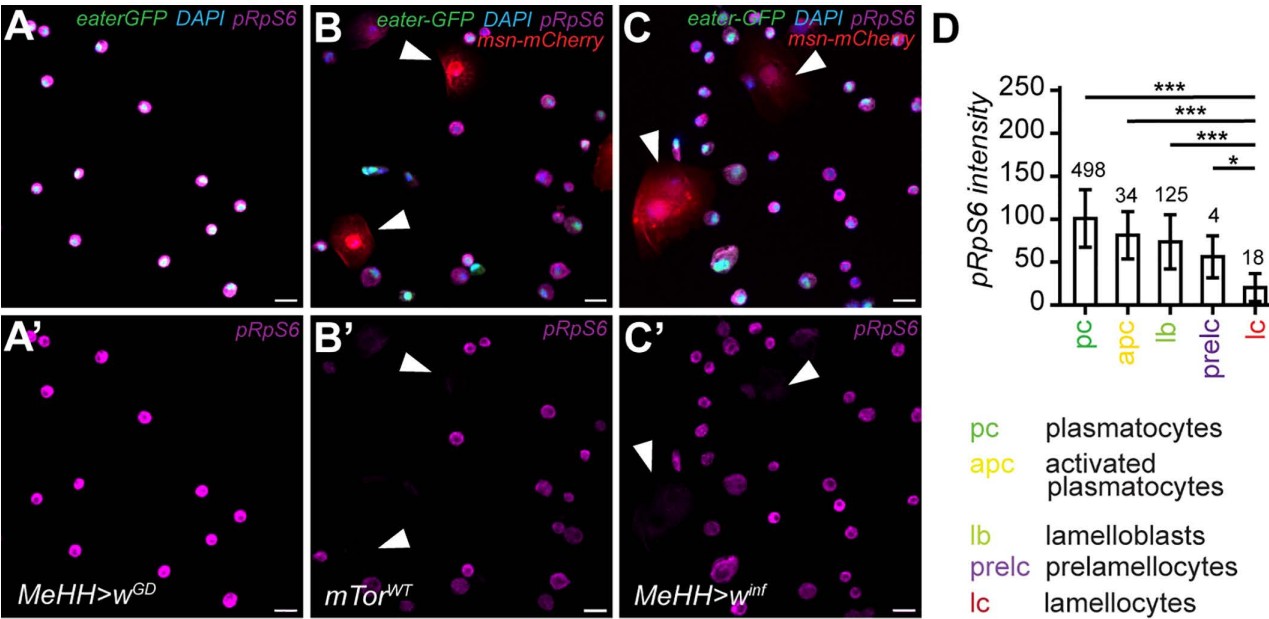

**Fig 6. mTORC1 signaling is downregulated in lamellocytes after *mTor^WT* overexpression and wasp infection.** A-C') Example images of pRpS6 immunostained hemocytes from uninfected (A-A' *MeHH>w^GD*), wasp-infected (B-B' *MeHH>w^GD inf*) and *mTor^WT* (C-C' *MeHH>mTor^WT*) late third instar larvae. Arrowheads point to two lamellocytes defined by *Msn-mCherry* expression. A-C) Maximum intensity projection of all channels (DAPI, GFP, mCherry, Alexa Fluor 647). A'-C') Alexa Fluor 647 channel separately depicting pRpS6 signal. D) Quantification of Alexa Fluor 647 staining intensity of hemocyte types pooled from all treatments (*MeHH>w^GD*, *MeHH>w^GD inf*, *MeHH>mTor^WT*). The quantification data are available in the S7 Table.

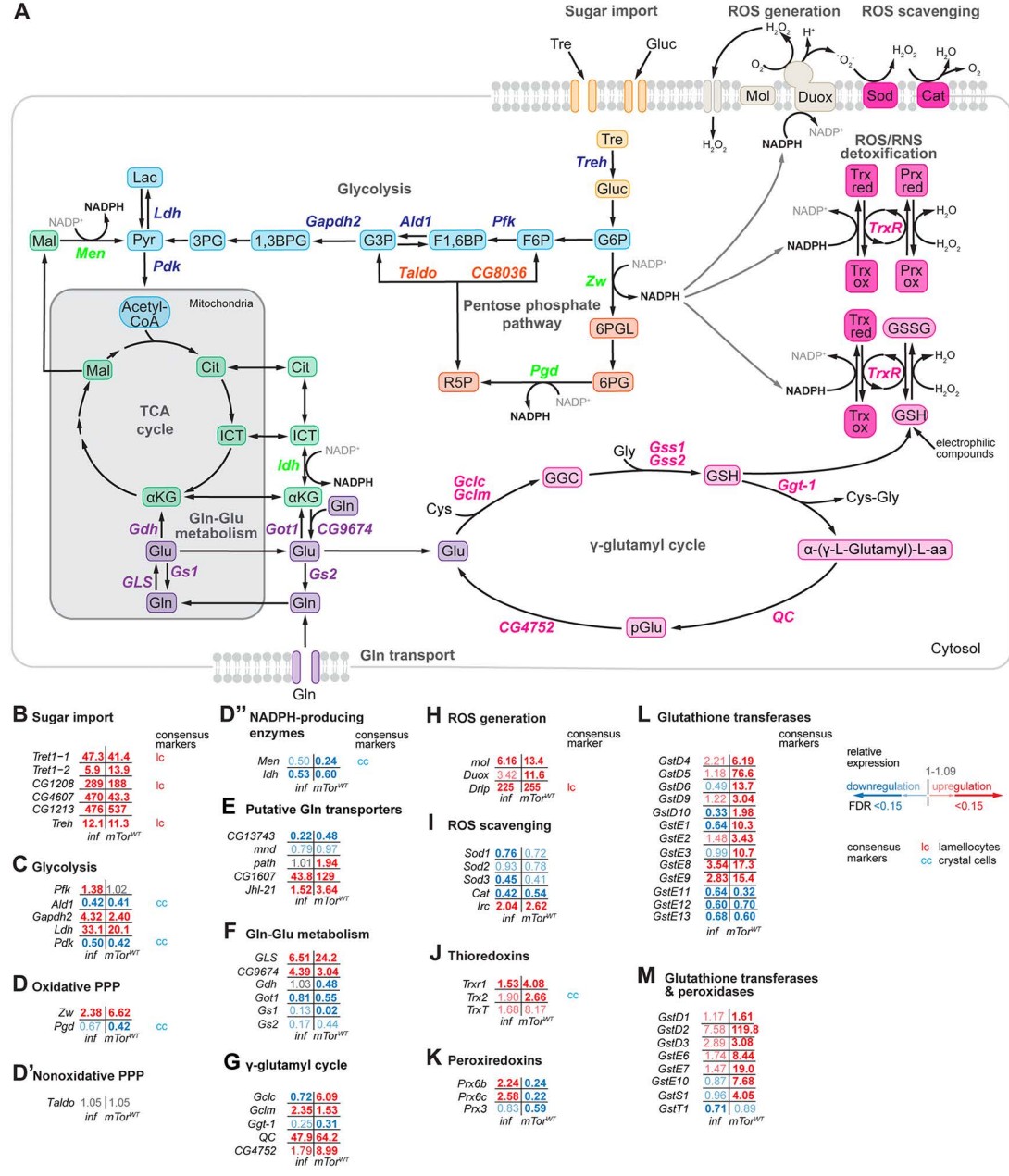

**Fig 7. *mTor^{WT}* overexpression and wasp infection alter the Redox metabolism in activated hemocytes.** A) Schematic diagrams of sugar import, glycolysis, oxidative pentose phosphate pathway (PPP), tricarboxylic acid (TCA) cycle, glutamine/glutamate metabolism, γ-glutamyl cycle, ROS generation and detoxification. Expression of selected genes involved in B) sugar import, C) glycolysis, D) oxidative PPP, D') non oxidative PPP, and D'') additional NADPH-producing enzymes, E) Glutamine/glutamate metabolism, F) γ-glutamyl cycle, G) ROS generation and H-M) ROS scavenging and detoxification. The gene expression data are available in the S4 Table. Abbreviations: Tre – Trehalose, Gluc – Glucose, G6P – Glucose-6-phosphate, F6P – Fructose-6-phosphate, F1,6 BP – Fructose-1,6-Bisphosphate, G3P – Glucose-3-phosphate, 1,3BPG – 1,3-Bisphosphoglycerate, 3PG – 3-phosphoglycerate, Pyr – Pyruvate, Lac – Lactate, Acetyl-CoA – Acetyl-Coenzyme A, Mal – malate, 6PGL – 6-phosphogluconolactone, 6PG – 6-Phosphogluconate, R5P – Ribulose-5-phosphate, Trx ox – oxidised Thioredoxin, Trx red – reduced Thioredoxin, TrxR – Thioredoxin Reductase, GSH – oxidised glutathione, GSSG – reduced glutathione, Prx ox – oxidised Peroxiredoxin, Prx red – reduced Peroxiredoxin, Glu – glutamate, Gln – glutamine, Cit – citrate, ICT – isocitrate, αKG – alpha-keto-glutarate, Cys – cystein, GGC – γ-glutamyl-L-cysteine, Gly – glycine, α-(γ-L-Glutamyl)-L-aa – alpha-(gamma-L-Glutamyl)-L-amino acid, pGlu – pyro-Glutamate (5-oxoproline). *Treh – Trehalase, Pfk – Phosphofructokinase, Ald1 – Aldolase 1, Gadph2 – Glyceraldehyde 3 phosphate dehydrogenase 2, Ldh – Lactate dehydrogenase, Pdk – Pyruvate dehydrogenase kinase, Men – Malic enzyme, Pgd – Phosphogluconate dehydrogenase, Zw – Zwischenferment, Idh – Isocitrate dehydrogenase, GLS – Glutaminase, Gdh – Glutamate*

dehydrogenase, Got1 – Glutamate oxaloacetate transaminase 1, Gs1 – Glutamine synthetase 1, Gs2 – Glutamine synthetase 2, CG9674 – Glutamate synthase (NADH), Gclc – Glutamate-cysteine ligase catalytic subunit, Gclm – Glutamate-cysteine ligase modifier subunit, Gss1 – Glutathione synthetase 1, Gss2 – Glutathione synthetase 2, Ggt-1 – γ-glutamyl transpeptidase, QC – Glutaminyl cyclase, CG4752 – 5-oxoprolinase (ATP-hydrolyzing), Mol – Moladietz, Duox – Dual oxidase, Sod – superoxide dismutase, Cat – catalase.

dehydrogenase (*Pgd*; Fig 7A, D), *Malic enzyme* (*Men*) and *Isocitrate dehydrogenase* (*Idh*; Fig 7A, D") were downregulated. These results imply that the PPP is a critical source of NADPH in activated hemocytes. Kazek et al. 2024 [63] showed that especially lamellocytes rely on cyclic PPP to sustain high NADPH production. In activated blood cells, when the demand is increased, glutamine (Gln) becomes a conditionally essential amino acid [64]. Gln is used as a source of energy to fuel the tricarboxylic acid (TCA) cycle and as a building block for nucleotide synthesis as well as for the synthesis of the potent antioxidant glutathione [65,66]. In mammals, several protein families have been implicated in Gln transport such as solute carrier proteins (SLC1,6,7 and 38 families) and Na+-Dependent Transporters (SLC5 family [67]). In *Drosophila* the SLC38 family ortholog *CG13743*, *minidiscs* (*mnd*) and *pathetic* (*path*, SLC36 family member) are implicated in Gln transport. *path*, which was previously shown to regulate growth via ILS/mTOR signaling [68], was two-fold upregulated by *mTor^WT* overexpression. Furthermore, *CG1607* and *Juvenile hormone Inducible-21* (*Jhl-21*), members of the SLC7 family and, thus putative Gln transporters, were induced in our data (Fig 7E, S4 Table). Gln can also be recycled through autophagy [69]. Autophagy-related genes were partially upregulated in our data (S11A' Fig, S3 Table).

During glutaminolysis, Gln is deaminated into glutamate (Glu) by Glutaminase (GLS) and processed further into alpha-Ketoglutarate (αKG) by Glutamate dehydrogenase (Gdh) to boost the TCA cycle for energy production. *GLS* was upregulated after wasp infection and *mTor^WT* overexpression, whereas *Gdh* was downregulated in *mTor^WT* hemocytes. Under normal growth conditions, Glu can be synthesized from αKG and Gln by an NAD-dependent glutamate synthase (CG9674). *CG9674* was upregulated in *mTor^WT* and wasp-infected hemocytes. Gln can be synthesized from Glu by the glutamate synthetases Gs1 and Gs2. *Gs1* was downregulated in *mTor^WT* and wasp-infected hemocytes (Fig 7A, F). Glu is essential for glutathione (GSH) synthesis by the γ-glutamyl cycle (Fig 7A, G). GSH is considered to be the main antioxidant in cells [70] and is enriched in hemocytes after wasp infection [63]. The catalytic subunit of *Glutamate-cystein ligase* (*Gclc*) was upregulated by both treatments while the modifier subunit (*Gclm*) was only upregulated after *mTor^WT* overexpression. The enzymes involved in recycling Glu and maintaining the glutathione pool, *glutaminyl-peptide cyclotransferase* (*QC*) and *CG4752*, the *5-oxo-prolinase*, were also upregulated (Fig 7A, G).

Overall, both stimuli shift hemocyte gene expression toward a metabolically activated profile, marked by increased expression of genes associated with glucose uptake, oxidative pentose phosphate pathway activity and glutamine-dependent glutamate and glutathione biosynthesis. These transcriptional signatures suggest a metabolic state that could support the increased energetic, biosynthetic, and redox requirements of activated hemocytes.

## Activated hemocytes show gene expression patterns consistent with NADPH-dependent ROS metabolism

The upregulation of the ROS-producing NADPH oxidase *Dual oxidase* (*Duox*) and its maturation factor *Moladietz* (*Mol*) after wasp infection and *mTor^WT* overexpression (Fig 7A, H) supports the idea that ROS production in activated hemocytes may be dependent on NADPH supplied by the PPP. However, both genes were quite lowly expressed in general (S4 Table). The upregulation of *mol* is critical for localizing Duox to and maintaining it at the cell membrane to enhance the rate and specificity of ROS production [71,72]. Excessive ROS is detrimental to the cell, and several antioxidant systems exist for ROS scavenging [73,74]. Superoxide ($O_2^-$), a byproduct of NADPH oxidases, is converted to hydrogen peroxide ($H_2O_2$) by superoxide dismutases (SODs [74]). The SODs were not significantly affected or were mostly downregulated (Fig 7A, I). Alternatively, $H_2O_2$ is converted to $H_2O$ by cellular antioxidant proteins such as catalases (Cat), thioredoxins (Trx [73,74]), peroxiredoxins (Prx) and Glutathione S-transferases (GSTs). *Cat* was downregulated while *immune-regulated catalase* (*Irc*) was upregulated under both conditions (Fig 7A, I). *Trxs* tended to be upregulated by wasp

infection and *mTor^WT* overexpression (Fig 7A, J). *Peroxiredoxin 6b* (*Prx6b*) and *Peroxiredoxin 6c* (*Prx6c*) were upregulated by wasp-infection but downregulated in *mTor^WT* overexpressing hemocytes (Fig 7A, K) suggesting that *PRxs* respond to genuine immune challenges and have similar cytoprotective functions as described for mammalian macrophages [75–77]. In contrast, GSTs were predominantly induced by *mTor^WT* overexpression (Fig 7A, L-M). This may account for the necessity to detoxify cytotoxic radicals in the absence of a genuine threat. Taken together, our data imply that activated hemocytes rely on aerobic glycolysis for energy production and increased glutamine-glutamate metabolism for glutathione synthesis. Importantly, the metabolic shifts inferred by gene expressional changes induced by wasp infection also take place after *mTor^WT* overexpression, suggesting that mTOR is an important energy rheostat in *Drosophila* hemocytes.

### *mTor* signals upstream of the JNK and the p38b pathways in lamellocyte hematopoiesis

mTOR intersects several signaling pathways such as PI3K/AKT1, JAK-STAT, MAPK1 and NF-κB in mammalian immune cells, ultimately leading to immune cell proliferation, differentiation and survival [19,78–80]. Thus, we surveyed our sequencing data for differentially regulated genes within signaling pathways known to be involved in mammalian and *Drosophila* blood cell activation, proliferation and differentiation with a focus on the JAK/STAT, JNK and p38 pathways (Fig 8A). Wasp infection and *mTor^WT* overexpression were sufficient to induce the ligands of the canonical JAK/STAT (*unpaired 1–3* (u*pd1–3*), Fig 8A'), the TNFα-Eiger (*eiger* (*egr*), Fig 8A'), the Pvr (*PDGF- and VEGF-related factor 2* (*Pvf2*), Fig 8A'), the EGFR (*gurken* (*grk*)), the FGFR (*thisbe* (*ths*), *pyramus* (*pyr*)), and the wingless (*Wnt oncogene analog 6* (*Wnt6*)) pathway (S5 Table). Interestingly, *Hedgehog* (*Hh*) was not expressed in hemocytes while other Hedgehog pathway components were, suggesting that Hh is produced elsewhere. The core components and the negative as well as the positive regulators of these pathways were well expressed in hemocytes, however, their regulation was not always consistent. The Toll and imd/Rel pathways were active after wasp infection and *mTor^WT* overexpression. However, inconsistencies between treatments, such as stronger induction of AMPs, were likely due to contaminating bacteria during oviposition (Fig 8A'-A"', S5 Table). To test whether the JAK-STAT, JNK and p38 pathways, known to induce lamellocytes [44,45] are involved in mTOR-mediated lamellocyte hematopoiesis, we took a genetic approach by expressing *mTor^WT* and simultaneously blocking the JAK/STAT, JNK and p38 pathways in hemocytes with dominant-negative or RNAi constructs. All constructs, except for *dome^DN*, reduced total cell counts in comparison to *mTor^WT* overexpression (blue boxes) to their respective control levels (pink boxes; Figs 8B; S14A). Silencing the JAK/STAT, JNK and p38b pathways reduced plasmatocyte lineage counts (plasmatocytes + activated plasmatocytes) to wild-type levels (Fig 8B'). However, in the lamellocyte lineage (lamelloblasts + prelamellocytes + lamellocytes) only silencing the JNK and p38 pathways reduced the lamellocyte counts to control levels downstream of *mTor^WT* (Fig 8B"). Thus, the JNK and the p38 pathways are engaged by mTOR signaling to initiate lamellocyte hematopoiesis. While the activation of JAK-STAT signaling is not required in hemocytes downstream of *mTor^WT* for this response, it might still be required upstream of mTor in hemocytes and in other tissues such as muscle [81,82] to induce lamellocyte fate (Fig 8B"').

## Discussion

mTOR signaling in immune cell differentiation and function has been predominantly investigated in mammalian hematopoietic stem cells (HSCs) [80], lymphoid [17,83] and myeloid cells [19]. In *Drosophila* hematopoiesis, mTOR has so far only been studied in the context of hemocyte progenitors in the lymph gland [46,47]. Here, for the first time, we genetically dissect the role of the endogenous mTOR pathway in peripheral hemocytes and show that it is involved in the activation of plasmatocytes and the differentiation of lamellocytes. Fig 9 depicts our model for mTOR-mediated reprogramming of metabolic gene expression, hemocyte activation and differentiation (Fig 9).

Genetic overexpression and wasp infection induce signaling pathways that recruit mTOR to induce reprogramming of metabolism and cell differentiation patterns leading to plasmatocyte activation and lamellocyte hematopoiesis. Both mTORC1 and mTORC2 are required for lamellocyte differentiation mediated by the Jnk and p38 pathways, however,

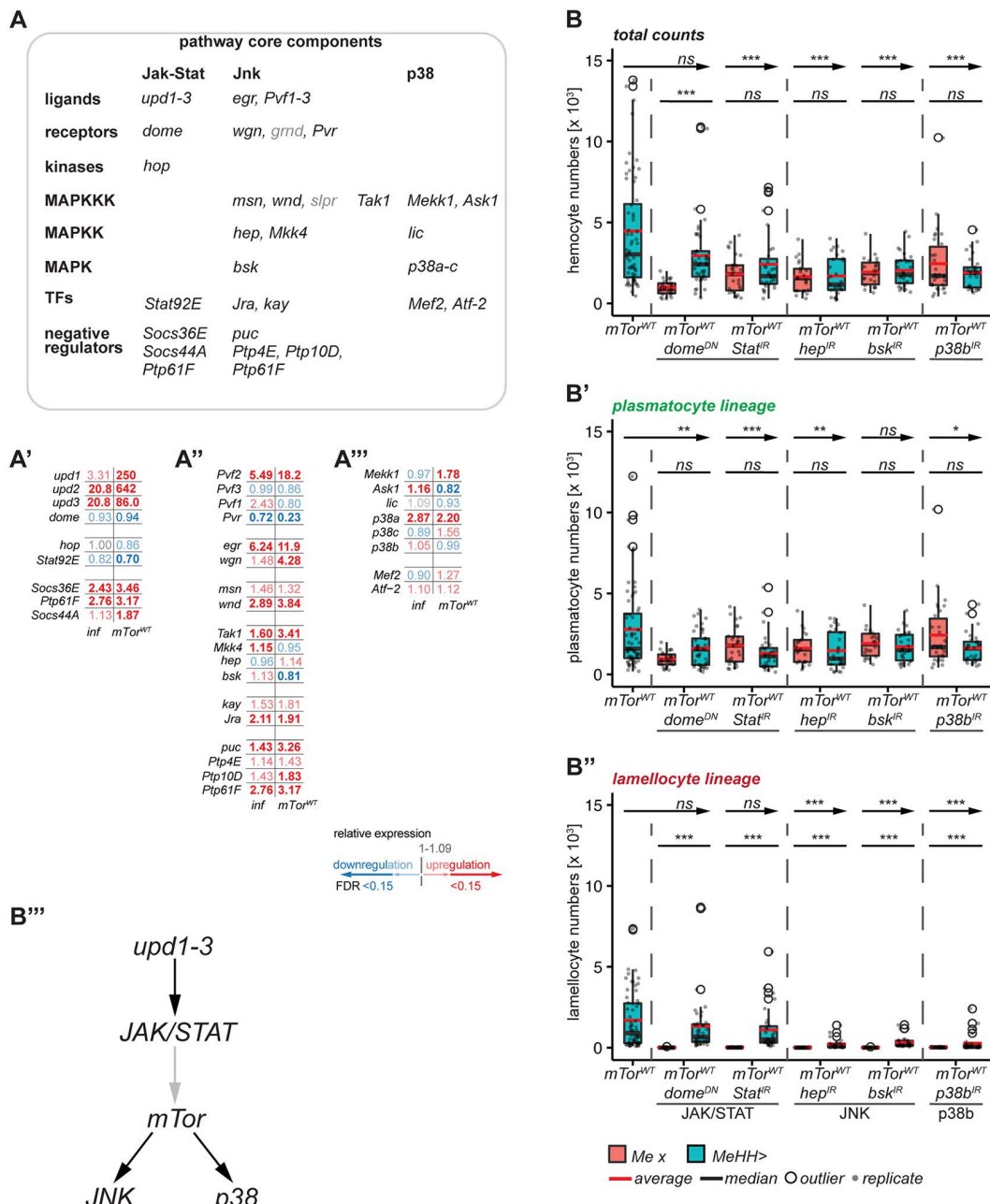

**Fig 8. *mTor* signals upstream of the JNK and the p38b pathways and downstream or in parallel with the JAK-STAT pathway in lamellocyte hematopoiesis.** A) Schematic of signaling modules for the JAK-STAT, JNK and p38b pathways. Of note, *grindelwald* (*grnd*) and *slipper* (*slpr*) were not expressed in hemocytes. RNA sequencing data for the depicted genes of A') the JAK-STAT pathway; A") the JNK pathway; A'") RNA sequencing data for the p38 pathway. B) Total hemocyte counts (n = 22–64) of *mTor^WT^* overexpression alone (*MeHH > mTor^WT^*) and *mTor^WT^* overexpression with simultaneous dominant negative or RNAi constructs for the JAK/STAT (*mTor^WT^ dome^DN^*, *mTor^WT^ Stat^IR^*), the JNK (*mTor^WT^ hep^IR^*, *mTor^WT^ bsk^IR^*) and the p38b (*mTor^WT^ p38b^IR^*) pathways (control: *Me x* – pink boxplots; transgene expression: *MeHH >*- blue boxplots); B') Cell counts of plasmatocyte lineage; B") Cell counts of lamellocyte lineage. The *MeHH > mTor^WT^* hemocyte counts are re-plotted from Figs 1D-D'; B'") Model for pathway interactions of the indicated pathways for *mTor^WT^* overexpression. Significance levels: *** p < 0.0001, ** p < 0.001, * p < 0.05, ns – not significant, arrow – statistical comparison of *MeHH > mTor^WT^* to *MeHH >*, line – statistical comparison of *Me x* to *MeHH >*. The gene expression data are available in the S5 Table and the hemocyte count data are available in S7 Table.

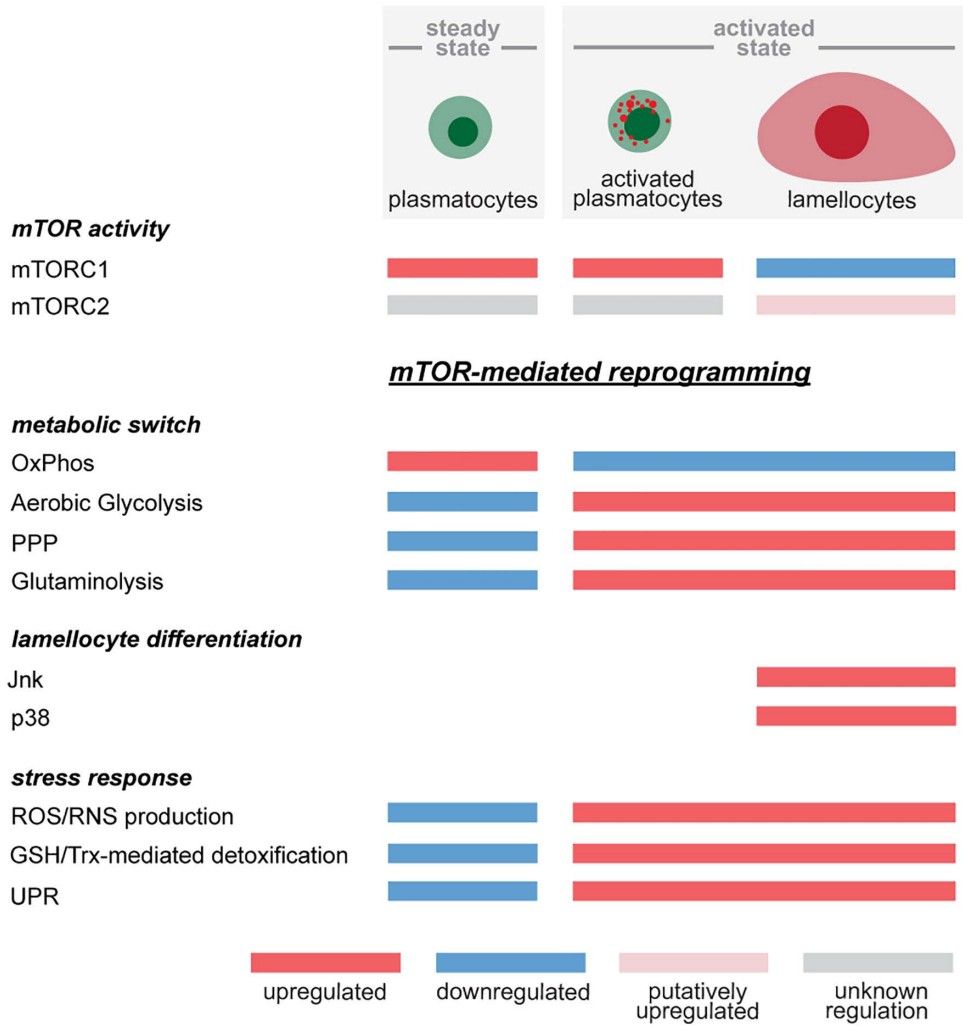

**Fig 9. Model for mTOR-mediated reprogramming of gene expression patterns associated with metabolism, hemocyte activation and differentiation.**

mTORC1 is downregulated in mature lamellocytes. The contribution of mTORC2 in all cell types is unclear, but data from mammalian cells suggest that mTORC2 regulates features important for lamellocyte function such as cytoskeletal rearrangement. The metabolic switch from steady state to activated hemocytes includes the downregulation of genes associated with OXPHOS and the upregulation of genes connected with aerobic glycolysis, PPP and glutaminolysis. The metabolic switch and the programming of cell activation and differentiation patterns result in the upregulation of stress response genes comprising ROS/RNS production, GSH/Trx-mediated detoxification and the UPR. It is not yet clear how the activated cell types contribute to the stress response.

The reduced killing phenotype after parasitoid wasp infection and the reduced lamellocyte numbers in the *raptor-rictor* double knock-down support the requirement for mTORC1 and mTORC2 signaling for hemocyte activation and differentiation after natural infection. Comparably, both mTOR complexes are needed for mammalian immune cell hematopoiesis and function [20,84–87]. However, selectively activating mTORC1 by inhibiting TSC led to an increase in round plasmatocyte-like cells (eater-GFP^high) among steady-state hemocytes, but did not promote lamellocyte formation.

mTORC1 may also play a role in maintaining hemocyte progenitor populations. In mammals, RAPTOR is essential for HSC self-renewal, and its loss leads to stem cell depletion and premature differentiation [88]. Consistent with this, *raptor* RNAi increased the abundance of lamelloblasts, cells we classified as lamellocyte precursors based on their approximately tenfold lower *eater-GFP* expression and smaller size relative to plasmatocytes [33]. However, it is possible that additional progenitor populations, such as prohemocytes, may also exhibit low eater-GFP levels and small size, making them indistinguishable from lamelloblasts in our analyses. The scarcity of molecular markers for defined cell types, particularly progenitors, is limiting the progress in understanding *Drosophila* peripheral hematopoiesis also in general. Thus, raptor may be required for progenitor cell maintenance, but it is uncertain for which progenitor type and whether there are different progenitor cell types in the periphery of *Drosophila* larvae.

To advance our understanding of how mTOR signaling contributes to blood cell activation, we examined the transcriptomes of uninfected, wasp infected and *mTor^WT* overexpressing peripheral larval hemocytes. We show that parasitoid wasp infection and *mTor^WT* overexpression cause largely overlapping changes in hemocyte gene expression patterns. Consequently, the similarities in gene expression profiles are due to the presence of similar hemocyte types after wasp infection and *mTor^WT* overexpression but might also be brought on by similar changes in mTOR signaling after hemocyte activation. Our RNA sequencing and experimental data indicate the downregulation of mTORC1 in lamellocytes regardless of whether lamellocytes are induced genetically or by wasp infection. Initially, mTORC1 signaling promotes the proliferation of all plasmatocyte-like cell types, as increased cell division is required for the demand-adapted hematopoiesis that follows wasp infection, including the expansion of lamellocyte precursors [33]. However, mTORC1 activation also drives terminal differentiation [86,89], and its activity subsequently declines in fully differentiated cells, which no longer divide [33,90]. As a result, mTORC1 becomes downregulated in lamellocytes, whereas it remains active in plasmatocytes, which retain the capacity to differentiate into additional cell types.

The suppression of mTORC1-related processes, even after *mTor^WT* overexpression, seems contrary to what would be expected from continuous *mTor^WT* expression. This may reflect a paradoxical reduction in signaling output, as excessive *mTor^WT* can sequester or dilute essential cofactors and impair mTOR complex assembly, as previously reported [48]. Our data collectively show that *mTor^WT* overexpression in hemocytes strongly alters metabolism-associated transcriptional patterns. Despite these pronounced effects, constitutive *mTor^WT* overexpression throughout development did not reduce adult fly lifespan (S15 Fig). Notably, the induction of stress-response genes, including heat-shock proteins and GSTs, was unique for *mTor^WT* overexpression. This observation aligns with the known role of endogenous mTOR in activating stress-responsive transcription factors, which drive the expression of genes involved in resistance to heat and ROS, among other stressors [91,92]. Particularly Hsp70 family members are known to block mTORC1 expression [93] and could be an additional mechanism by which mTOR limits its own excessive expression and detrimental effects. Thus, collectively, our data imply that mTORC1 suppression is a characteristic of lamellocyte differentiation and function. This is in agreement with the differential regulation of mTORC1 activity throughout the life cycle of various myeloid and lymphoid cell types required for cell fate decisions and adequate cell function [18,23,94,95]. Future work, such as manipulation of *raptor* or *Tsc1/gig* in lamellocytes after wasp infection, is needed to directly assess whether mTORC1 activity in differentiated lamellocytes is required for their effector functions.

Processes regulated by mTORC2, such as cytoskeletal rearrangements [96,97] and increased sugar transport alongside aerobic glycolysis [12,98,99], were also upregulated in our dataset. These changes are consistent with known lamellocyte functions [63,100,101]. Thus, although mTORC1 appears to be downregulated, mTORC2 likely remains active in lamellocytes to regulate lamellocyte-specific features. However, due to the lack of reliable molecular tools for assessing mTORC2 activity in *Drosophila*, we were unable to experimentally confirm its involvement in lamellocytes or other activated hemocyte populations.

The metabolic switch from mitochondrial OXPHOS to cytosolic aerobic glycolysis is a well characterized mTOR-mediated response to increased energy and biosynthetic demands in mammalian immune [102–105] and cancer cells

[106–108]. Increased glucose intake and activation of aerobic glycolysis accompany hemocyte activation also in *Drosophila* [1,60,61,63]. In mammals, mTOR increases sugar uptake and glycolysis by transcriptional and translational control of Hif-1α [109]. *Sima,* the *Drosophila* homolog of *Hif-1a,* is involved in regulating glycolytic gene expression [60,110,111]. Despite *mTor^WT* overexpression and upregulation of glycolytic genes such as *Ldh*, *sima* transcription was not increased and its binding partner *tgo* was downregulated. Intriguingly, Gli-similar *sugarbabe* (*sug*), a transcription factor known to also regulate carbohydrate metabolism and genes involved in glycolysis in *Drosophila* [112] was 7-fold upregulated in *mTor^WT* overexpressing hemocytes. It remains an open question why only *mTor^WT* overexpressing hemocytes induce *sug*, but not wasp-activated hemocytes.

Glucose can also be utilized in the oxidative pentose phosphate pathway (PPP) to generate NADPH for detoxification and production of ROS [113]. For instance, neutrophiles utilize PPP-produced NADPH to facilitate the oxidative burst needed for their immune-effector function [114]. Kazek et al. [63] showed increased flow of glucose through the oxidative PPP in activated hemocytes after wasp infection. Unlike us, they did not observe any major transcriptional changes in PPP genes, however, knockdown of *Zw* in hemocytes reduced lamellocyte differentiation and drastically impaired the survival after parasitoid wasp infection [63]. PPP activation can therefore be considered as an important metabolic adaptation in lamellocytes. Moreover, PPP pathway activation was also indicated in hemocytes in the medullary zone of the lymph gland [39] and during the shift from embryonic to larval hemocytes [34].

The role of Gln-Glu metabolism is poorly characterized in hemocyte activation. Kazek et al. [63] showed by tracing the incorporation of $^{13}$C from $^{13}$C-labeled glucose into Gln during the early course of infection (16 hours *post infection* + 6 hours $^{13}$C-glucose feeding) that Gln and its derivatives fuel the TCA cycle regardless of whether the hemocytes were sourced from parasitoid infected or uninfected larvae. Our data, taken at a comparatively late time point (48 hours *post infection*), point towards Gln and Glu being processed through the y-glutamyl cycle to produce glutathione in activated hemocytes, purportedly to alleviate oxidative stress. The utilization of Gln and Glu might be time point specific. Early after infection, these amino acids might contribute to energy production by mitochondria to fuel proliferation, however, later, they might be involved in the antioxidative stress response. More research is required to determine the temporospatial pattern of Gln-Glu utilization during the immune activation.

ROS has so far been linked to plasmatocytes [115–117], however, the involvement of other hemocyte types in ROS production, distribution and detoxification is not yet well understood. Killing of the parasitoid wasp eggs and larvae requires $H_2O_2$ which is further metabolized into cytotoxic hydroxyl radicals [115,118]. Our data indicate $H_2O_2$ production through the upregulation of *Duox* and its maturation factor *mol* in activated hemocytes. However, in our dataset and in previously published transcriptomic analyses [35,63] *Duox* and *mol* gene expression levels in hemocytes remain comparatively low, even following wasp parasitization. Nevertheless, some experimental evidence supports a functional role for the Duox system in hemocyte-derived ROS production. Kinoshita et al. [119] reported upregulation of Duox in hemocytes of the *mxc^mbn1* hematopoietic tumor model, accompanied by reduced ROS levels upon Duox silencing. The context-dependent contribution of the Duox system to ROS generation in activated hemocytes warrants further experimental investigation.

The aquaporin *Drip*, a lamellocyte marker [34,40], was highly upregulated after wasp infection and in *mTor^WT* overexpressing hemocytes. Aquaporins, in addition to facilitating the diffusion of $H_2O$ across cellular membranes, are also permeable to $H_2O_2$ [120,121]. Hemocytes of adult flies acquire intracellular $H_2O_2$ via the aquaporin Prip at wound sites, where it is required for hemocyte activation and *Upd3* induction [122]. Drip channels DUOX-produced $H_2O_2$ back into the cytoplasm of *Drosophila* neurons where it inhibits dendritic growth [123]. Our data put forward the idea that activated hemocytes, possibly lamellocytes, are the source of $H_2O_2$ and that Drip functions as an $H_2O_2$ channel to facilitate ROS distribution. NADPH oxidases are dormant in resting phagocytic cells and activated only by a stimulus such as infection [124]. If lamellocytes were the main source of $H_2O_2$ production during a parasitoid infection, the inducibility of lamellocytes might be a mechanism of protecting the host against damage by cytotoxic radicals. Dolezal and colleagues, however,

suggested that ROS may be produced by plasmatocytes and crystal cells and lamellocytes take over a protective role by forming the outer layer of the capsule scavenging and detoxifying ROS [125].

For the first time, our data provide a connection between the requirement for the JNK and p38 pathway for lamellocyte hematopoiesis downstream of mTor. The JNK and p38 pathways play a pivotal role in the response to oxidative stress in *Drosophila* [126–128]. Intriguingly, they also control lamellocyte hematopoiesis [44,45]. Our data also show the upregulation of other immune- and stress-related signaling pathways previously linked to lamellocyte hematopoiesis such as the Toll pathway [129] as well as stress responsive biosynthetic pathways such as GSH biosynthesis. Coincidentally, the regulation of the GSH synthesis is also under the transcriptional control of Nuclear response factor 1/2 (Nrf1/2), the JNK, p38 and Toll pathways [130]. The connection between lamellocyte hematopoiesis and function, ROS production and ROS detoxification as well as mTOR signaling is an interesting topic for future studies.

Emerging evidence from mammalian innate immune cells and leukocytes underscores the pivotal role for mTOR in connecting environmental and cellular stimuli with metabolism to control proliferation, differentiation and survival. Our data implicate similar functions for mTOR in *Drosophila* hemocytes. Proper timing and location of mTOR activation play an essential role in the outcome of the activation. The temporal dynamics of mTORC1 and mTORC2 for proper immune cell function is currently not well understood in any study system. However, the *Drosophila* hematopoietic system offers a feasible and an easily tractable genetic model for further research of immune signaling networks involving mTOR.

## Materials and methods

### Fly stocks and fly husbandry

A detailed description of fly lines and their genotypes for each experiment are in S6 Table. In summary, a combined double GAL4-driver, *Hml$^\Delta$>EGFP* and *He > GFP$^{nls}$* (hereafter *HH>* [129]) with partially overlapping specificity was used to express UAS-constructs in hemocytes for melanotic nodule scoring and parasitoid wasp assays. *Hml$^\Delta$-GAL4* is expressed in all plasmatocytes [131] while *He-GAL4* is expressed in all hemocyte classes but only in 80% of all hemocytes [44]. We used GD, KK and SH RNAi lines from the Vienna *Drosophila* Resource Center (VDRC) and their corresponding genetic background strains denoted as *w$^{GD}$*, *w$^{KK}$* and *w$^{SH}$* as wild type controls and *mTor$^{IR}$* from the TRiP collection with its corresponding background (S6 Table).

For flow cytometric hemocyte counting as well as imaging hemocytes and larvae, the hemocyte reporters (gifts from R. Schulz) *eater-GFP$^{nls}$* (specific for plasmatocytes [132]) and *MsnF9mo-mCherry* (specific for lamellocytes [133]) were combined with *Hml$^\Delta$-GAL4* and *He-GAL4* to create "*MeHH>*". The recombined hemocyte reporter line *yw, eater-GFP, MsnF9mo-mCherry* ("*Me*" [33]) was used as control. The original *UAS-mTor$^{WT}$* contained a *hsFlp* construct on the second chromosome which was crossed out prior to the experiments.

To determine the epistatic interactions of the Jak/Stat, the JNK and the p38b pathways with *mTor*, we created compound fly stocks of *UAS-mTor$^{WT}$* with RNAi constructs for the transcription factor *Stat92E* (*UAS-mTor$^{WT}$ UAS-Stat$^{IR}$*), the Jun kinase kinase *hemipterous* (*UAS-mTor$^{WT}$ UAS-hep$^{IR}$*), the Jun kinase *basket* (*UAS-mTor$^{WT}$ UAS-bsk$^{IR}$*) as well as the *p38b* kinase (*UAS-mTor$^{WT}$ UAS-p38b$^{IR}$*). The compound stock of *UAS-mTor$^{WT}$* with a dominant negative construct for the JAK-STAT receptor *domeless* (*UAS-mTor$^{WT}$ UAS-dome$^{DN}$*) was created by Hairu Yang. Before creating new fly lines, *UAS-mTor$^{WT}$*, *UAS-Tor$^{DN}$*, *UAS-dome$^{DN}$*, *UAS-hep$^{IR}$*, *UAS-p38b$^{IR}$* were backcrossed six times into the *w$^{1118}$* genetic background line of the VDRC GD library, *w$^{GD}$*.

Flies were reared on food containing 36 g mashed potato powder, 9 g agar, 45.5 ml corn syrup, 14.5 g dry yeast, 8 g nipagin and 5 g ascorbic acid per 1 l of water. 15–20 virgin females were crossed to 10 males and kept at 25 °C in a 12:12 h light:dark cycle. Experimental flies were transferred into fresh vials daily, and vials containing eggs were transferred to 29 °C. Larvae were analyzed mainly at the late third instar or as indicated. All fly crosses are listed in S6 Table.

## Melanotic nodules

Third instar larvae were placed individually in drops of water on a 12-well hydrophobic printed slide (Thermo Scientific). The presence of melanotic nodules was examined using a stereomicroscope. Examples of larvae with and without nodules in Fig 1A were imaged using a Nikon DS-Vi1 digital camera (Nikon Metrology Inc., MI, U.S.A) and Nikon SMZ/745T stereomicroscope (Nikon Instruments Inc., NY, U.S.A). 60–70 larvae of four replicate crosses were analyzed.

## Parasitoid wasp encapsulation assay

15 female and 10 male *Leptopilina boulardi G486* parasitoid wasps were placed into a vial with second instar fly larvae. After 2 hours, the wasps were removed, and the vials were placed back to 29 °C for 48 hours. To score the encapsulation response, fly larvae were placed individually in drops of water on a 12-well hydrophobic printed slide (Thermo Scientific) and dissected with forceps using a stereomicroscope. Encapsulation was judged unsuccessful when a living, unmelanized wasp larva alone or with remnants of a melanized capsule was found. In a successful reaction, a wasp egg or larva was encapsulated and melanized. In general, most of the examined larvae harbored only one wasp larva, but occasional multiparasitization occurred. When both melanized and unmelanized wasp larvae were found inside the *Drosophila* larva, the response was categorized unsuccessful. The wasp assays were replicated three times for each genotype, and 50–100 infected larvae were assayed for each replicate depending on experiment.

## Reverse transcription quantitative PCR (RT-qPCR) on hemocytes

Third instar larvae were gently washed several times in filtered water and individually dissected with forceps in a drop of ice-cold 1 x phosphate-buffered saline (PBS) on a 12-well slide. Hemocytes from 60 larvae were pooled per sample. Each sample was spun down for 7 minutes at 2500 x g at 4 °C. Excess PBS was removed (leaving some behind because the hemocyte pellet was often invisible), hemocytes were frozen on dry ice and stored at − 80 °C. Hemocyte RNA was extracted using a Norgen single Cell RNA purification kit (Norgen Biotek Corp, Ontario, Canada) including DNase treatment (RNase free Dnase I kit, Norgen). The RNA quality and concentration was analyzed with a NanoDrop ND-1000 spectrophotometer (Thermo Scientific). The SYBR Green real-time one-step qPCR kit (Thermo Fisher Scientific) reaction setup was as follows: 5 μl iTaq universal SYBR Green reaction mix (2x), 0.125 μl iScript reverse transcriptase, 0.3 μl of 10 μM forward and reverse primers, 2.275 μl Nuclease-free water and 2 μl of 10 ng/μl RNA. Reactions were run with a BioRad CFX96 system with the following protocol: reverse transcription reaction at 50 °C for 10 minutes, polymerase activation and denaturation at 95 °C for 1 minutes followed by 36 amplification cycles at 95 °C with denaturation for 10 seconds and annealing/extension at 60 °C for 15 seconds. At the end, a melt curve analysis was performed in 0.5 °C increments between 65–95 °C, 2 seconds for each step. NRT (no reverse transcriptase) and NT (no template) controls were included in each run for quality control. Each experiment consisted of three biological replicates with two to three technical replicates. The expression ratios of target-to-reference gene (*mitochondrial ribosomal protein S24*, *mRpS24*) in controls and treatment groups were calculated using the dCT-method ($-2^{\Delta dCT}$). Primer sequences were as follows: *mRpS24*: 5'-TCAGGATTCCACACCAGCTC-3' and 5'-GCCAATAAAATGCGGCGGAT-3'; *mTor*: 5'- CCCATAAAACTCTGGTGATGC-3' and 5'-AACTGCTCGTAGGCTTCCTG-3'.

## Flow cytometry

Late third instar larvae were gently washed several times in filtered water with a brush and individually placed in 20 μl of cold 8% bovine serum albumin (BSA) in 1 x PBS on a 12-well slide. Larvae were dissected open at full length on the dorsal side to release the hemolymph. Carcasses were removed and the drop containing hemolymph was pipetted into 1.5 ml Eppendorf vials with 80 μl of 8% BSA in PBS. The hemolymph samples were analyzed within one hour after dissections with a BD Accuri C6 flow cytometer (Becton Dickinson). The full protocol for hemocyte flow cytometry is described elsewhere [33]. In short, hemocytes were gated on a side scatter-forward scatter plot (SSC-A/FSC-A) to separate them

from debris. The location of the hemocyte population was verified by back-gating *eater-GFP-positive* hemocytes to the SSC-A/FSC-A plot. Non-fluorescent, *eater-GFP-positive* and *Msn-mCherry-positive* hemocytes were used as single-color controls to locate the position of the *GFP-only*, the *mCherry-only* and the double-positive populations on a FL1-A (GFP)-FL3-A (*mCherry*) plot. A 488 nm 50 mW solid-state laser was used to excite the fluorochromes. A 510 ± 15 nm (FL1) and a 610 ± 20 nm (FL3) optical filters were used for fluorophore detection. To correct fluorescence spill over, 8% of the GFP signal was deducted from FL3. We classified hemocytes into two lineages and five populations based on the fluorescence signal as described [33]. The plasmatocyte lineage includes hemocytes expressing high levels of *eater-GFP*: plasmatocytes (*GFP*<sup>high</sup>) and activated plasmatocytes (*GFP*<sup>high</sup>, *mCherrry*<sup>low</sup>). The lamellocyte lineage is characterized by low or no *eater-GFP* and increasing *Msn-mCherry* expression: lamelloblasts (*GFP*<sup>low</sup>, *mCherrry*<sup>neg</sup>), prelamellocytes (*GFP*<sup>low</sup>, *mCherrry*<sup>low</sup>) and mature lamellocytes (*GFP*<sup>neg</sup>, *mCherrry*<sup>high</sup>). Hemocyte samples were collected from approximately 10 larvae in three replicate crosses for each genotype. 30 µl of each sample was analyzed and the results were multiplied by 3.3333 for a total hemocyte load.

### Fluorescence imaging of *Drosophila* larvae

Late third instar larvae were gently washed several times in filtered water with a brush, dried and placed in a drop of 65% glycerol on a 76 x 26 mm microscope glass slide (Thermo Scientific) and covered with a 20 x 20 mm #1 coverslip (Gerhard Menzel GMBH, Braunschweig, Germany). The slides were kept at 4 °C overnight to completely immobilize the larvae for imaging. Larvae were imaged with a 5x air objective of a Zeiss Axio Imager 2 microscope equipped with an ApoTome.2. Seven *MeHH > Tor*<sup>WT</sup> and three *MeHH > w*<sup>GD</sup> larvae were imaged.

### Hemocyte samples and RNA extraction for RNA sequencing

Third instar larvae were washed several times in filtered water and individually dissected with forceps in drops of ice-cold 1 x PBS on a 12-well slide. Hemocyte samples were collected from developmentally matched uninfected control (*HH x w*<sup>GD</sup>) and hemocyte-directed *mTor*<sup>WT</sup>-overexpression (*HH > mTor*<sup>WT</sup>) larvae as well as infected control (*HH > w*<sup>GD</sup> *inf*) and hemocyte-directed *mTor*<sup>WT</sup>-overxpression (*HH > mTor*<sup>WT</sup> *inf*) larvae 48 h after *L. boulardi* infection. Hemolymph samples were collected on ice in cohorts of approximately 30 larvae. Samples were immediately spun down for 7 minutes at 2500 x g at 4 °C, excess PBS was removed and hemocytes were frozen on dry ice. Hemocytes of 100 larvae were pooled for each of the three replicate RNA samples at the RNA extraction step. RNA was extracted using a Norgen single Cell RNA purification kit (Norgen Biotek Corp, Ontario, Canada) including DNase treatment (RNase free Dnase I kit, Norgen). The RNA quality and concentration was analyzed with a NanoDrop ND-1000 spectrophotometer (Thermo Scientific) and with a Fragment analyzer. Samples were stored at -80 °C.

### RNA sequencing and data analysis

Bulk RNA sequencing was carried out at the Finnish Functional Genomics Centre (Turku, Finland). Samples were prepared with an Illumina TrueSeq Stranded mRNA sample Preparation kit and sequenced using single-end sequencing and 50 bp read length with a HiSeq 3000 instrument. Base calling was done with the Bcl2fastq2 version 1.8.4 software (Illumina). Reads were aligned to the *Drosophila* reference genome (dm6) using Star version 2.5.0 and the number of reads associated with each gene was counted using the *subreads* package version 1.5.0. Gene-wise read counts were normalized using the TMM method in the R/Bioconductor package. The normalized read counts were transformed into counts per million (CPM) for statistical analyses to consider the sequencing depth. The whole dataset is available at GEO's Omnibus genome database with a reference number GSE237065. Data was log10 transformed and pairwise comparisons were analyzed with 2-tailed t-tests. The Holm-Hochberg method was used to calculate the false discovery rate (FDR). Genes with no/very low expression, defined by less than a total of 10 reads across the replicates within a treatment, were removed. Differential expression with an FDR < 0.15 was considered significant. In addition, when filtering

treatment-specific genes (in wasp-infected *only* or in *mTor^WT*-overexpressing *only*), the FDR value was set to > 0.5 in the treatment group where the gene was considered not differentially expressed. Gene Ontology (GO) analysis was performed using FlyMine (version 53, February 2022 [134]).

## Ribosomal protein S6 kinase (S6k) immunoblotting

Hemocytes from 300 third instar larvae per genotype were collected by dissecting the larvae in ice-cold Ringer's solution (130 mM NaCl, 5.0 mM KCl, 1.0 mM $CaCl_2$) containing a phosphatase and protease inhibitor cocktail (Roche PhosSTOP tablet and Roche cOmplete tablet, EDTA free) in DMSO in a final concentration of 1:1000. Hemocytes were pelleted by centrifugation at 2500 x g for 7 minutes at 4 °C and stored at -80 °C. 80 µl lysis buffer (50 mM Tris-HCl pH 8.0, 150 nM NaCl, 1% Non-idet P-40, 0.5% Sodium deoxycholate, 0.1% SDS) with the addition of 1 mM PMSF and 1:1000 phosphatase and protease inhibitor solution in DMSO was added to the frozen samples, followed by incubation on ice for 60 minutes and centrifugation at 12000 x g for 10 minutes at 4 °C. The supernatant was transferred into fresh vials and the protein concentration was measured using a BCA assay kit (Merck Millipore) with a standard curve and absorbance at 562 nm according to manufacturer's instructions. 35 µg of protein was loaded onto a Bio-Rad Criterion TGX 7.5% precast gel and run at 70 V for 2 hours inProSieve EX Running buffer. Proteins were transferred onto a Midi 0.2 µm nitrocellulose membrane using the Bio-Rad Trans-blot Turbo Transfer system and the Mixed MW program designed for a transfer of proteins ranging from 5 - 150 kDa. Blots were blocked with 1% fish gelatin for 45 minutes at room temperature while shaking. An antibody against *Drosophila* S6k protein sequence CKEHIQEGIV (MW 1155.34, purity 98.46) was manufactured at ProteoGenix. The peptide was conjugated to a keyhole limpet hemocyanin (KLH) carrier and affinity purified against the antigen. An anti-ATP5A antibody (Abcam #ab14748) was used as a loading control (1:50000). S6k protein specificity was tested using wild type, S6k knockdown and S6k overexpressing larval hemocytes (S12A Fig). Deviations from the protocol described below are stated in the S12 Fig caption. To detect phosphorylation at mTORC1 phosphorylation site Threonine 398 in the S6k protein, a *Phospho-Drosophila* p70 S6k ($T_{398}$; Cell Signaling Technology #9209) antibody was used. The anti-S6k antibodies were diluted 1:1000 in 5% BSA in 1 x TBS-Tween, as recommended in the Cell Signaling protocol. Overnight incubation at + 4 °C with rotation was followed by five times 5 minute washes with 1 x PBS-Tween at room temperature while shaking. Secondary antibodies (Vectorlab Goat Anti-Rabbit IgG Antibody (H + L), Peroxidase P-1000 and Goat Anti-Mouse IgG Antibody (H + L), Peroxidase P-2000) were diluted 1:10000 in 2.5% fat-free milk in 1 x PBS-Tween. 1:15000 dilution of StrepTactin and horseradish peroxidase (HRP) conjugate was used for the detection of Bio-Rad Precision Plus Protein WesternC unstained standards. After 2 hours of incubation at room temperature while shaking, blots were washed 4 x for 5 minutes with 1 x PBS-Tween followed by a final wash with 1 x PBS for 5 minutes. Proteins were detected with SuperSignal West Femto Maximum Sensitivity Substrate (Thermo Fisher) and imaged with a Bio-Rad ChemiDoc imager.

## Phospho-Ribosomal protein S6 (pRpS6) immunostaining

Third instar larvae were washed and dissected on 12-well glass slides in 20 µl of cold 1% BSA in PBS with a phosphatase and protease inhibitor cocktail (final concentration 1:1000), 2–3 larvae per well. Hemocytes were let to attach on the slides for 40 minutes at room temperature. After fixing in 20 µl of 10% formalin, hemocytes were rinsed three times with ice-cold 1 x PBS followed by 5 minutes permeabilization with 0.1% Triton X-100. After careful washing and rinsing with PBS, samples were pre-blocked with 5% BSA in PBS for 20 minutes and stained with 1:200 dilution (in 5% BSA in PBS) of an antibody against *Drosophila* phospho-RpS6 (Ser233/235/239) [135]. After 1 hour, the hemocytes were again washed three times with 1 x PBS and stained with goat anti-rabbit secondary antibody (Alexa Fluor 647; 1:500 in 5% BSA in PBS) for 1 hour. Hemocytes were washed three times with 1 x PBS and mounted with ProLong Diamond Antifade mountant with DAPI (Invitrogen) and imaged using a Zeiss LSM 780 laser scanning confocal microscope mounted on a Cell Observer microscope with a Plan Apochromat 63 x/1.4 oil immersion objective. Z-stacks were taken in 0.83 µm increments. The

laser lines used were the pulsed diode laser 405 nm for DAPI, multiline Argon laser 488 nm for GFP, diode laser 561 nm for mCherry and InTune-tunable pulsed laser 628 nm for Alexa Fluor 647. Images were processed using ImageJ2 version 2.9.0/1.53t. First, Z-projections of the Alexa Fluor 647 channel were made by summing up the pixel values along the slices, and images were rendered into 8-bit images. Regions of interest (ROIs) were determined as cell areas based on *GFP* and *mCherry* expression and mean gray values in the Z-projected Alexa Fluor 647 channel were obtained for each ROI in the image. Hemocytes were classified as discussed above. In addition, cell shape and size were used in the classification. Plasmatocytes and lamelloblasts were separated from each other by the level of the gray value of the GFP intensity (plasmatocyte: gray value of GFP intensity > 30; lamelloblast: gray value of GFP intensity < 30) based on the GFP intensities in the control samples, where the majority of hemocytes were plasmatocytes.

### Life span

Experimental crosses were kept at 25 °C and on day one after egg lay flies were transferred into new vials or discarded and vials containing eggs were placed at 29 °C for subsequent development. Eclosing flies were collected on three consecutive days, separated by sex and placed into fresh vials (six to eight vials per cross per sex, 13–22 flies per vial). Flies were transferred onto fresh food on every third day, and at the same time dead individuals were counted.

### Data analysis

The data were plotted and analyzed using the R versions 3.6.0 (2019-04-26) 4.3.1 (2023-06-16). ggplot2 was used for plotting the data [136]. The melanotic nodule prevalence and encapsulation response was analyzed with logistic regression with binomial distribution in the lm4 package [137]. The multcomp package was used for pairwise comparisons [138]. The hemocyte count data was overdispersed and were analyzed by applying a negative binomial generalized linear model or a zero-inflated model if zero values were overrepresented using the MASS package [139]. The least square means (estimated marginal means) were analyzed for multiple comparisons, and the Tukey method was used for adjusting the p-value using the emmeans package [140]. The RT-qPCR and pS6k/S6k ratio data were first checked for normal distribution and equal variances and were then analyzed using a linear model (ANOVA) with the car package [141] and with Tukey's method for pairwise comparisons in the multcomp package. When needed, log transformation was applied to data prior to analysis. The pRpS immunostaining data was analyzed using the non-parametric Kruskal-Wallis test due to heteroscedasticity, followed by Wilcoxon rank sum test for pairwise comparisons. The Benjamini-Hochberg method was used to correct for multiple comparisons. Principal component analysis (PCA) was conducted and plotted with Bioconductor's PCAtools package [142]. For the statistical analysis of the fly life span, Log-Rank (Mantel-Cox) test was applied using the GraphPad Prism 9 software.

### Supporting information

**S1 Table. RNA sequencing data, and DE genes shared between *mTor^{WT}* overexpressing hemocytes and hemocytes from wasp infected larvae as well as DE genes specific to each of the treatments (Fig 4).**
(XLSX)

**S2 Table. Gene Ontology (GO) terms (Fig S8C).**
(XLSX)

**S3 Table. GO terms + genes of mTORC1-controlled processes (Fig 5).**
(XLSX)

**S4 Table. Metabolism-related genes (Fig 7).**
(XLSX)

**S5 Table. Selected Signaling pathways previously shown to be involved in mammalian hematopoiesis (Fig 8).**
(XLSX)

**S6 Table. Acquired and created fly stocks and fly crosses.**
(XLSX)

**S7 Table. Data files (Fig 1–3; Fig 5; Fig 8; S2-S6 Fig; S12 Fig; S13 Fig; S15 Fig).**
(XLSX)

**S1 Fig. Representative scatterplots of hemocyte populations uninfected and infected late L3 larvae shown in the indicated Figs.** A) *MeHH>w^{GD}*, *MeHH>mTor^{WT}* and *MeHH>mTor^{DN}* (Fig 1). B) *Me>Tsc1^{KK110811}; gig^{GD6313}*, *MeHH>Tsc1^{KK110811}; gig^{GD6313}* (Fig 2). C) *Me x raptor^{KK106491},rictor^{SH33079}*, *MeHH x raptor^{KK106491},rictor^{SH33079}* (Fig 3).
(TIF)

**S2 Fig. Hemocyte phenotypes of *mTor^{WT}* larvae with (+) or without (-) melanotic nodules.** A) *In vivo* hemocyte phenotype of control (*MeHH>w^{GD}*), *eaterGFP* representing the plasmatocyte lineage and *MsnCherry* the lamellocyte lineage; A') *in vivo* hemocyte phenotype of *mTor^{WT}* hemocyte-directed overexpression larvae without nodules (*MeHH>mTor^{WT}*-); A"-A"') *in vivo* hemocyte phenotypes of *mTor^{WT}* hemocyte-directed overexpression larvae with nodules (*MeHH>mTor^{WT}*+). Red asterisk – lymph glands, red arrowheads – segmental peripheral plasmatocytes, white arrowheads – segmental muscles, blue arrowheads – pharyngeal muscles, red V – alary muscle fibres, white arrow – lamellocyte, magenta V – pericardial cells, red arrow – nodule, scale bars 500 μm. B) Total hemocyte counts of control (*MeHH>w^{GD}*; n = 31) and *mTor^{WT}* overexpression in hemocytes (without nodules *MeHH>mTor^{WT}*-; n = 44, with nodules *MeHH>mTor^{WT}*+; n = 20); B') Cell counts of plasmatocyte lineage; B") Cell counts of lamellocyte lineage. To facilitate plotting, we removed the highest outlier for *MeHH>mTor^{WT}*+ (5950 lamelloblasts). The data presented in B-B" is identical with Fig 1D-D" but separately plotted for larvae with and without nodules. Significance levels: *** $p < 0.0001$, ** $p < 0.001$, * $p < 0.05$, ns – not significant. The hemocyte count data are available in the S7 Table.
(TIF)

**S3 Fig. *mTor* RNAi phenocopies the *mTor^{DN}* construct.** A) Total hemocyte counts of uninfected control (*MeHH>attP^{2}684A*, n = 30) and *mTor* RNAi (*MeHH>mTor^{IR}*, n = 30); A') Cell counts of plasmatocyte lineage; A") Cell counts of lamellocyte lineage. B) Total hemocyte counts of infected control (*MeHH>attP^{2}684A*, n = 30) and *mTor* RNAi (*MeHH>mTor^{IR}*, n = 30); B') Cell counts of plasmatocyte lineage; B") Cell counts of lamellocyte lineage. Significance levels: *** $p < 0.0001$, ** $p < 0.001$, * $p < 0.05$, ns – not significant. The hemocyte count data are available in the S7 Table.
(TIF)

**S4 Fig. Loss of *Tsc1* in hemocytes does not phenocopy lamellocyte hematopoiesis nor the encapsulation response.** A) Total hemocyte counts (n = 30) of uninfected control (*MeHH>w^{KK}*, n = 30) and *Tsc1* RNAi larvae (*MeHH>Tsc1^{KK}*, n = 59); A') Cell counts of plasmatocyte and A") lamellocyte lineage. B) Total counts of infected control (*MeHH>w^{KK}*, n = 30) and *Tsc1* RNAi larvae (*MeHH>Tsc1^{KK}*, n = 55); B') Cell counts of plasmatocyte and B") lamellocyte lineage. C) *Tsc1* RNAi (*HH>Tsc1^{GD}*, n = 316; *HH>Tsc1^{KK}*, n = 301) and controls (*w x Tsc1^{GD}*, n = 316; *w x Tsc1^{KK}*, n = 301). Significance levels: *** $p < 0.0001$, ** $p < 0.001$, * $p < 0.05$, ns – not significant. The hemocyte count and wasp encapsulation data are available in the S7 Table.
(TIF)

**S5 Fig Loss of *gigas (gig)* does not phenocopy lamellocyte hematopoiesis but enhances cellular immunity against *L. boulardi* infection.** A) Total hemocyte counts of uninfected control (*MeHH>w^{GD}*, *MeHH>w^{KK}*, n = 30) and *gig* RNAi larvae (*MeHH>gig^{GD}*, n = 52; *MeHH>gig^{KK}*, n = 43); A') Cell counts of plasmatocyte lineage; A") Cell counts of lamellocyte lineage. B) Total counts of wasp infected control (*MeHH>w^{GD}*, *MeHH>w^{KK}*,n = 30) and *gig* RNAi larvae

($MeHH>gig^{GD}$, n=61; $MeHH>gig^{KK}$, n=17); B') Cell counts of plasmatocyte lineage; B") Cell counts of lamellocyte lineage. C) Parasitoid wasp assay of control ($w \times gig^{GD}$, n=154) and *gig* RNAi larvae ($HH>gig^{GD}$, n=152; $HH>gig^{KK}$ was lethal). Significance levels: *** $p<0.0001$, ** $p<0.001$, * $p<0.05$, *ns – not significant*. The hemocyte count and wasp encapsulation data are available in the S7 Table.
(TIF)

**S6 Fig. Neither *raptor* nor *rictor* RNAi alone block lamellocyte hematopoiesis nor the immune response to parasitoid wasp infection.** A) Total hemocyte counts (n=25–31) of uninfected controls ($MeHH>w^{GD}$, $MeHH>w^{KK}$, $MeHH>w^{SH}$), *raptor* ($MeHH>raptor^{GD}$, $MeHH>raptor^{KK}$) and *rictor* ($MeHH>rictor^{SH}$) RNAi; A') Cell counts of plasmatocyte; A") Cell counts of lamellocyte lineage. B) Total hemocyte counts (n=22–30) of infected controls ($MeHH>w^{GD}$, $MeHH>w^{KK}$, $MeHH>w^{SH}$), *raptor* ($MeHH>raptor^{GD}$, $MeHH>raptor^{KK}$) and *rictor* ($MeHH>rictor^{SH}$) RNAi; B') Cell counts of plasmatocyte; B") Cell counts of lamellocyte lineage. C) Encapsulation assay of *raptor* ($HH>raptor^{GD}$, n=306; $HH>raptor^{KK}$, n=301) and *rictor* RNAi ($HH>rictor^{SH}$, n=158) and their respective controls ($w \times raptor^{GD}$, n=319; $w \times raptor^{KK}$, n=302; $w \times rictor^{SH}$, n=141) in hemocytes. Significance levels: *** $p<0.0001$, ** $p<0.001$, * $p<0.05$, *ns – not significant*. The hemocyte count and wasp encapsulation data are available in the S7 Table.
(TIF)

**S7 Fig. Summary of the genetic experiments.** A) Hemocyte lineages (plasmatocyte lineage: plasmatocytes, activated plasmatocytes; lamellocyte lineage: lamelloblasts, prelamellocytes, lamellocytes), mTORC1 (*raptor*), mTORC2 (*rictor*) and TSC (*Tsc1*, *gig*); A') hemocyte-directed expression of $mTor^{DN}$, $mTor^{IR}$, $raptor^{IR}$ and $raptor^{IR}$,$rictor^{IR}$ leads to blocking of mTOR signaling (blue) and, in general, to increase in lamelloblasts (marked with "+") with minor effects on other hemocytes types, while hemocyte-directed expression of $mTor^{WT}$ and $Tsc1IR$,$gig^{IR}$ leads to the activation of mTOR signaling (red), and to activation of hemocytes including lamellocyte differentiation; A") wasp-infection in wild type larvae ($w^{GD}$) reduces plasmatocytes and induces the formation of lamelloblasts, activated plasmatocytes, prelamellocytes and lamellocytes; A"') expression of $mTor^{DN}$, $mTor^{IR}$, $raptor^{IR}$ and $raptor^{IR}$,$rictor^{IR}$ in hemocytes of wasp-infected larvae reduces the plasmatocyte lineage hemocytes, and in case of $mTor^{DN}$ and $mTor^{IR}$ also increases lamelloblast and prelamellocyte numbers. Expression of $mTor^{WT}$ and $Tsc1IR$,$gig^{IR}$ have only minor effects in addition to wasp infection. The hemocyte schematics are adapted from Anderl *et al.* [33].
(TIF)

**S8 Fig. Expression of consensus hemocyte markers and GO term analysis.** A) Consensus markers for lamellocytes, crystal cells and plasmatocytes. Unlike the other crystal cell markers, *Heat shock protein 83* (*Hsp83*), *Glutathione S transferase E3* (*GstE*), *lazaro (laza)* and *CG34136* were upregulated in $mTor^{WT}$ hemocytes. *Notch* (*N*) was downregulated by wasp infection and $mTor^{WT}$ expression. As crystal cell markers, *Prophenoloxidase 1* (*PPO1*) and *Prophenoloxidase 2* (*PPO2*) were downregulated, while the lamellocyte-specific *Prophenoloxidase 3* (*PPO3*) was upregulated by both treatments. B) Markers for selected plasmatocyte subclusters. The "stress/GST" markers were identified both in the clusters PM5 [35] and GST [36], the "mitotic" markers were enriched in PL-prolif [34] and X [39], and the "AMP" genes in at least two of the clusters PL-AMP [34] PM7 [35], AMP [38] and PH6 [39], respectively. C) Representative GO terms enriched in the three groups of genes that are highlighted as green, blue, or red in B. Gene expression data and hemocyte markers are available in the S1 Table and the GO terms in the S2 Table.
(TIF)

**S9 Fig. mTor upstream activators.** A) Schematic of mTOR complexes and mTOR activating pathways such as amino acid-activated and growth factor-activated mTORC1 signaling as well as Insulin-like signaling (ILS)-activated mTORC2 signaling. B) Expression patterns of mTORC1 (*mTor*, *raptor*, *Telo2 interacting protein 1* (*Tti*), *Lst8*, *Proline-rich Akt substrate 40 kDa* (*PRAS40*)) and C2 components (*mTor*, *rictor*, *Tti1*, *Lst8*, *SAPK-interacting protein 1* (*Sin1*)).

Experimental data in *Drosophila* suggests that Lst8 is not part of mTORC1, but only of mTORC2 [143]. PRAS40 is thought to affect mTOR activity only in ovaries [144] but is expressed in hemocytes (S1 Table); B') Expression patterns of genes involved in amino acid sensing. The *D. melanogaster* amino acid sensors comprise *Sestrin* (*Sesn*), which senses methionine, leucine and other branched chain amino acids [145,146], and the S-adenosylmethionine sensor *Unmet expectations* (*unmet*; [147]). By binding to their cognate amino acids, the sensors release their inhibition on the GATOR2 complex (grey inhibitory symbol). GATOR2 (*Missing oocyte* (*mio*), *WD repeat domain 24* (*Wdr24*), *WD repeat domain 59* (*Wdr59*), *Nucleoporin at 44A* (*Nup44A*), *Secretory 13* (*Sec13*)) sequesters and thereby blocks the inhibitory function of GATOR1 (Nitrogen permease regulator-like 2 *Nprl2*, Nitrogen permease regulator-like 3 *Nprl3*, Increased minichromosome loss 1 *Iml1*) on the Ragulator Complex (grey inhibitor symbol, RagA-B, RagC-D and Ragulator (*Late endosomal/lysosomal adaptor, MAPK and MTOR activator 1* (*Lamtor1*), *Late endosomal/lysosomal adaptor, MAPK and MTOR activator 3* (*Lamtor3*)). The *D. melanogaster* homolog of *Folliculin*, *Birt-Hogg-Dube* (*BHD*) is involved in leucine sensing [148]; B") Expression patterns of a growth factor antagonist (*argos* (*aos*)), of growth factors (gurken (*grk*), Keren (*Krn*), *spitz* (spi), *Pvf2*) and growth factor processing enzymes (*Rhomboid intramembrane proteases* (*rho*, *rho 4–7*)). The associated RTKs *EGFR* and *Pvr* were not differentially expressed; B"') Expression patterns of Insulin-like signaling (ILS) pathway components. ILS antagonist *Edysone-inducible gene L2* (*ImpL2; [149]*), *Insulin-like peptide 8* (*Ilp8*), *Phosphatidylinositol 3-kinase 92E* (Pi3K92E), negative regulator *Phosphatase and tensin homolog* (*Pten*) and *Akt kinase*. The Insulin-like receptor (*InR*) was not differentially expressed and *Ilp8* was the only differentially expressed *Ilp*. Consensus markers – marker genes for main hemocytes classes based on Hultmark & Ando, 2022 [29]. The gene expression data are available in S1 Table.
(TIF)

**S10 Fig. Mechanisms to block mTORC1.** A) Schematic of AMPK-mediated suppression of mTORC1 and gene expression patterns of involved genes; A') Schematic REDD1-mediated repression of mTORC1 and gene expression patterns of *scyl* and *charybde* (*chrb*); A") Schematic of p53-mediated suppression of mTORC1 and gene expression patterns of *septin interacting protein 3* (*sip3*), *Companion of reaper* (*Corp*) and *p53*. The gene expression data are available in S1 Table.
(TIF)

**S11 Fig. Cellular processes activated and blocked by mTORC1.** A) Schematic mTORC1-mediated phosphorylation of transcription factors activating protein, nucleotide and lipid synthesis, glucose and energy metabolism as well as gene expression patterns of transcription factors and effector genes in these processes; A') Schematic of mTORC1-mediated repression of apoptosis, autophagy and lysosome biogenesis as well as gene expression patterns of key genes of these processes. Consensus markers – marker genes for main hemocytes classes based on Hultmark & Ando, 2022 [29]. The gene expression data are available in S1 Table.
(TIF)

**S12 Fig. Western blot analysis of S6k and pS6k in hemocytes.** A) To test the newly generated total S6k antibody, we used *S6k* knockdown hemocytes (*HH > S6k$^{GD}$*) as a negative control and overexpression of S6$k^{CA}$ (*HH > S6k$^{CA}$*) as a positive control. Hemocytes from 200 larvae were pooled per genotype and 30 µg of protein was loaded into each well. In S6$k^{CA}$, Threonine at $T_{398}$ is replaced by glutamic acid rendering the site at least partially independent of upstream mTOR activation. ATP5 was used as a loading control. As evident from the blot, the S6k protein levels were very low in control hemocytes. The amount of S6k decreased to virtually undetectable levels in the *S6k* knock-down and increased in *S6k* overexpressing hemocytes, verifying that the antibody is detecting S6k *in vivo*. B) pS6k and S6k in hemocytes with *mTor$^{WT}$* overexpression and in hemocytes from wasp-infected larvae. Even though we increased the pools of larvae to 300, and loaded 35 µg of protein into each well, the levels of pS6k were in general very low in all samples, again especially in the uninfected control hemocytes. A bulk larval hemocyte collection method would be required to have enough protein for proper detection of S6k protein and phospho-protein in untreated hemocyte samples; B') Ratio of pS6k to total S6k in

hemocytes. the ratio of the pS6k band intensity of three blots to one blot of the total S6k of the same samples. *ns – not significant*. The data are available in the S7 Table.
(TIF)

**S13 Fig. pRps6 staining of hemocytes.** A) To verify the functionality of the pRps6 antibody in hemocytes, we knocked down *S6k* (*HH>S6k^GD^*) and checked the effect on the pRpS6 (Alexa Fluor 647) signal intensity. As expected, the pRpS6 staining intensity was reduced in *S6k* knock-down hemocytes. B) Example images of the pRpS6 staining in control (*MeHH>w^GD^*), wasp-infected (*MeHH>w^GD^* inf) and *mTor^WT^* overexpressing (*MeHH>mTor^WT^*) hemocytes. DAPI – nuclei, GFP – eaterGFP-positive hemocytes, mCherry – *MsnCherry*-positive hemocytes, Alexa Fluor 647 – pRpS6 staining. Arrowheads point to lamellocytes. The images here are the same as the maximum intensity projections in Fig 6A-C, but here all channels are shown separately. C) Quantification of the pRpS6 stain intensity in hemocytes from late third instar larvae. Significance levels: *** $p<0.0001$, ** $p<0.001$, * $p<0.05$. The data are available in the S7 Table.
(TIF)

**S14 Fig. Representative scatterplots of hemocyte populations uninfected and infected late L3 larvae shown in Fig 8.** A) *Me x mTor^WT^dome^DN^* and *MeHH>mTor^WT^dome^DN^*. B) *Me x mTor^WT^ Stat^IR^* and *MeHH>mTor^WT^ Stat^IR^*. C) *Me x mTor^WT^ hep^IR^* and *MeHH>mTor^WT^ hep^IR^; Me x mTor^WT^ bsk^IR^* and *MeHH>mTor^WT^ bsk^IR^*. D) *Me x mTor^WT^ p38b^IR^* and *MeHH>mTor^WT^ p38b^IR^*.
(TIF)

**S15 Fig. *mTor^WT^* overexpression in hemocytes does not reduce the lifespan of the flies.** Lifespan of *HH>w^GD^* (118 females and 116 males), *w x mTor^WT^* (108 females and 103 males) and *mTor^WT^*-overexpressing (*HH>mTor^WT^*, 120 females and 120 males) flies. Flies were maintained at 29 °C. Significance levels: *** $p<0.0001$, ** $p<0.001$, * $p<0.05$. The data are available in the S7 Table.
(TIF)

## Acknowledgments

We wish to thank emeritus professor Howard Jacobs for providing us with the total S6k antibody and Professor Aurelio Teleman for the pRpS6 antibody. We acknowledge the Biocenter of Finland, the Tampere *Drosophila* Facility and the Tampere Imaging Facility (TIF) for their service. We wish to thank the Bloomington Drosophila Stock Center and the Vienna Drosophila Research Center for fly stocks. We thank Emeritus Professor Dan Hultmark for his contribution to the RNA sequencing analysis and his valuable comments on the manuscript.

## Author contributions

**Conceptualization:** Ines Anderl, Laura Vesala.

**Data curation:** Ines Anderl, Laura Vesala.

**Formal analysis:** Ines Anderl, Laura Vesala.

**Funding acquisition:** Mika Rämet, Tiina Susanna Salminen, Laura Vesala.

**Investigation:** Ines Anderl, Jens-Ola Ekström, Tea Tuomela, Tiina Susanna Salminen.

**Methodology:** Ines Anderl, Laura Vesala.

**Project administration:** Ines Anderl, Laura Vesala.

**Resources:** Mika Rämet, Laura Vesala.

**Validation:** Laura Vesala.

**Visualization:** Ines Anderl, Laura Vesala.

**Writing – original draft:** Ines Anderl, Laura Vesala.

**Writing – review & editing:** Ines Anderl, Jens-Ola Ekström, Tea Tuomela, Mika Rämet, Tiina Susanna Salminen, Laura Vesala.

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
