## [Decision Letter · Decision Letter 0]

3 Feb 2026

PGENETICS-D-25-01328

mTOR signaling regulates demand-adapted hematopoiesis and metabolic reprogramming required for an effective cellular immune response in Drosophila melanogaster larvae

PLOS Genetics

Dear Dr. Vesala,

Thank you for submitting your manuscript to PLOS Genetics. After careful consideration, we feel that it has merit but does not fully meet PLOS Genetics's publication criteria as it currently stands. Therefore, we invite you to submit a revised version of the manuscript that addresses the points raised during the review process.

Please submit your revised manuscript within by Mar 05 2026 11:59PM. If you will need more time than this to complete your revisions, please reply to this message or contact the journal office at plosgenetics@plos.org. Please include the following items when submitting your revised manuscript:

We look forward to receiving your revised manuscript.

Kind regards,

Giovanni Bosco, Ph.D.

Section Editor

PLOS Genetics

Giovanni Bosco

Section Editor

PLOS Genetics

Aimée Dudley

Editor-in-Chief

PLOS Genetics

Anne Goriely

Editor-in-Chief

PLOS Genetics

**Journal Requirements:**

- TM on pages: 33, 37, and 38.

3) Some material included in your submission may be copyrighted. According to PLOSu2019s copyright policy, authors who use figures or other material (e.g., graphics, clipart, maps) from another author or copyright holder must demonstrate or obtain permission to publish this material under the Creative Commons Attribution 4.0 International (CC BY 4.0) License used by PLOS journals. Please closely review the details of PLOSu2019s copyright requirements here: PLOS Licenses and Copyright. If you need to request permissions from a copyright holder, you may use PLOS's Copyright Content Permission form.

Potential Copyright Issues:

i) Please confirm (a) that you are the photographer of 1A, or (b) provide written permission from the photographer to publish the photo(s) under our CC BY 4.0 license.

ii) Figure S7. Please confirm whether you drew the images / clip-art within the figure panels by hand. If you did not draw the images, please provide (a) a link to the source of the images or icons and their license / terms of use; or (b) written permission from the copyright holder to publish the images or icons under our CC BY 4.0 license. Alternatively, you may replace the images with open source alternatives. See these open source resources you may use to replace images / clip-art:

4) In the online submission form, you indicated that your data will be submitted to a repository upon acceptance. We strongly recommend all authors deposit their data before acceptance, as the process can be lengthy and hold up publication timelines. Please note that, though access restrictions are acceptable now, your entire minimal dataset will need to be made freely accessible if your manuscript is accepted for publication. This policy applies to all data except where public deposition would breach compliance with the protocol approved by your research ethics board. If you are unable to adhere to our open data policy, please kindly revise your statement to explain your reasoning and we will seek the editor's input on an exemption.

2) If any authors received a salary from any of your funders, please state which authors and which funders..

6)  Please ensure that the funders and grant numbers match between the Financial Disclosure field and the Funding Information tab in your submission form. Note that the funders must be provided in the same order in both places as well.

**Reviewers' comments:**

Reviewer's Responses to Questions

**Comments to the Authors:**

Reviewer #1: This study examines the role of mTOR in activating hemocytes to defend Drosophila larvae against parasitoids. This important study has been conducted thoroughly and carefully. The study is based on genetic manipulations specifically in hemocytes and analyzes the effects of these manipulations on hematopoiesis and gene expression in hemocytes. Similar to infection, mTOR overexpression leads to hemocyte activation and increased resistance, resulting in largely similar global changes in gene expression, including metabolic genes. Conversely, mTOR downregulation reduces activated plasmatocytes and resistance. This supports the authors' conclusion that mTOR is important for the metabolic switch in hemocytes that are activated to fight the parasitoid. However, analysis of expression after mTOR suppression and, in particular, direct analysis of metabolism is lacking, so the results clearly show that mTOR plays an important role in hemocyte activation and immune response, and although it is clear that mTOR is a key metabolic regulator, the results of this work did not directly test the role of mTOR in the metabolic switch of hemocytes. This is my only major criticism of the work. As part of the minor revision, I recommend toning down statements that draw strong conclusions about the effects on metabolism. In particular, I recommend emphasizing the limitations of this work. First, RNAseq suggests many things, but it is still far from analyzing metabolism. Second, I would emphasize more often that this is bulk RNAseq. The authors correctly emphasize this point in some passages. Therefore, conclusions about specific types of hemocytes are limited. The only analysis in this work that differentiates the effect on specific types of hemocytes is the analysis of S6k phosphorylation using an antibody directly in hemocytes.

Particular notes:

Page 21: “The other NADPH-producing enzymes Phosphogluconate dehydrogenase (Pgd; Fig. 7A, D), Malic enzyme (Men) and Isocitrate dehydrogenase (Idh; Fig. 7A, D’’) were downregulated implying that the main source of NADPH in activated hemocytes is Zw as shown earlier by Kazek et al. 2024 (63).” The upregulation of Zw and the downregulation of Pgd gene expression demonstrate the limitations of analyzing metabolism solely by mRNA expression. Both enzymes are required to complete the oxidative pentose phosphate pathway, so the statement that Zw is the main source of NADPH is misleading.

Page 29/30: “Thus, collectively, our data imply that mTORC1 suppression is a characteristic of lamellocyte differentiation and function.” This is a limitation of the study. Is the suppression of mTORC1 in differentiated lamellocytes important for their function? This was not tested, and this limitation should be mentioned in the discussion with the proposal of potential future experiments, such as activating mTORC1 specifically in lamellocytes upon infection.

Page 30: “The metabolic switch from mitochondrial OXPHOS to aerobic glycolysis observed in our data…” This is an example of overly strong phrasing. The actual metabolic switch was not observed in this work. This is an example of where the limitations of the study should be considered.

Page 31: “Kazek et al. (63) showed by 13C labeling of Gln during the early course of infection (16 hours pi + 6 h Gln 13C incubation)” Note: 13C-Gln was not used in the Kazek et al. study, but rather, glutamine was labeled in vivo from 13C-labeled glucose fed to the larvae. Glutamine metabolism was then traced based on its labeling pattern.

Page 32: “Our data indicate H2O2 production by upregulation of Duox and its cofactor moladietz in activated hemocytes.” ROS production in activated hemocytes remains questionable. Although Duox and moladietz expression increases in this study, their expression in hemocytes is extremely low based on their CPMs compared to other genes (STable4) — a finding that has been reported repeatedly in other studies using RNAseq on hemocytes. This calls into question the importance of Duox for ROS production in hemocytes.

Supporting data files with the data used to produce the figures' plots should be provided and specified in the figure legends.

**Have all data underlying the figures and results presented in the manuscript been provided?**

Large-scale datasets should be made available via a public repository as described in the *PLOS Genetics*
data availability policy , and numerical data that underlies graphs or summary statistics should be provided in spreadsheet form as supporting information., and numerical data that underlies graphs or summary statistics should be provided in spreadsheet form as supporting information.

Reviewer #1: **No:** Supporting data files with the data used to produce the figures' plots should be provided and specified in the figure legends.Supporting data files with the data used to produce the figures' plots should be provided and specified in the figure legends.

PLOS authors have the option to publish the peer review history of their article (what does this mean? ). If published, this will include your full peer review and any attached files.). If published, this will include your full peer review and any attached files.

**Do you want your identity to be public for this peer review?** For information about this choice, including consent withdrawal, please see our For information about this choice, including consent withdrawal, please see our Privacy Policy ..

Reviewer #1: **Yes:** Tomas DolezalTomas Dolezal

**Figure resubmission:**
---

## [Editor Report · Decision Letter 1]

11 Mar 2026

Dear Dr Vesala,

We are pleased to inform you that your manuscript entitled "mTOR signaling regulates demand-adapted hematopoiesis and metabolic reprogramming required for an effective cellular immune response in Drosophila melanogaster larvae" has been editorially accepted for publication in PLOS Genetics. Congratulations!

Yours sincerely,

Giovanni Bosco, Ph.D.

Section Editor

PLOS Genetics

Giovanni Bosco

Section Editor

PLOS Genetics

Aimée Dudley

Editor-in-Chief

PLOS Genetics

Anne Goriely

Editor-in-Chief

PLOS Genetics

BlueSky: @plos.bsky.social

Comments from the reviewers (if applicable):

**Data Deposition**

If you have submitted a Research Article or Front Matter that has associated data that are not suitable for deposition in a subject-specific public repository (such as GenBank or ArrayExpress), one way to make that data available is to deposit it in the Dryad Digital Repository . As you may recall, we ask all authors to agree to make data available; this is one way to achieve that. A full list of recommended repositories can be found on our . As you may recall, we ask all authors to agree to make data available; this is one way to achieve that. A full list of recommended repositories can be found on our website ..

http://datadryad.org/submit?journalID=pgenetics&manu=PGENETICS-D-25-01328R1

Additionally, please be aware that our data availability policy  requires that all numerical data underlying display items are included with the submission, and you will need to provide this before we can formally accept your manuscript, if not already present. requires that all numerical data underlying display items are included with the submission, and you will need to provide this before we can formally accept your manuscript, if not already present.

**Press Queries**

If you or your institution will be preparing press materials for this manuscript, or if you need to know your paper's publication date for media purposes, please inform the journal staff as soon as possible so that your submission can be scheduled accordingly. Your manuscript will remain under a strict press embargo until the publication date and time. This means an early version of your manuscript will not be published ahead of your final version. PLOS Genetics may also choose to issue a press release for your article. If there's anything the journal should know or you'd like more information, please get in touch via plosgenetics@plos.org ..

---

## [Editor Report · Acceptance letter]

PGENETICS-D-25-01328R1

mTOR signaling regulates demand-adapted hematopoiesis and metabolic reprogramming required for an effective cellular immune response in Drosophila melanogaster larvae

Dear Dr Vesala,

We are pleased to inform you that your manuscript entitled "mTOR signaling regulates demand-adapted hematopoiesis and metabolic reprogramming required for an effective cellular immune response in Drosophila melanogaster larvae" has been formally accepted for publication in PLOS Genetics! Your manuscript is now with our production department and you will be notified of the publication date in due course.

With kind regards,

Zsofia Freund

PLOS Genetics

On behalf of:
